# Measurement report: Leaf-scale gas exchange of atmospheric reactive trace species (NO₂, NO, O₃) at a northern hardwood forest in Michigan

Wei Wang[1], Laurens Ganzeveld[2], Samuel Rossabi[1], Jacques Hueber[1], Detlev Helmig[1]

[1]Institute of Arctic and Alpine Research, University of Colorado, Boulder, CO 80309, USA
[2]Wageningen University, Meteorology and Air Quality Section, Wageningen, the Netherlands

*Correspondence*: Wei Wang (wei.wang-3@Colorado.edu)

**Abstract.** During the **P**rogram for **R**esearch on **O**xidants: **PH**otochemistry, **E**missions, and **T**ransport (PROPHET) campaign from July 21 to August 3, 2016, field experiments of leaf-level trace gas exchange of nitric oxide (NO), nitrogen dioxide (NO₂), and ozone (O₃) were conducted for the first time on the native American tree species *Pinus strobus* (eastern white pine), *Acer rubrum* (red maple), *Populus grandidentata* (bigtooth aspen), and *Quercus rubra* (red oak) in a temperate hardwood forest in Michigan, USA. We measured the leaf-level trace gas exchange rates and investigated the existence of an NO₂ compensation point, hypothesized based on a comparison of a previously observed average diurnal cycle of NOₓ (NO₂ + NO) concentrations with that simulated using a multi-layer canopy exchange model. Known amounts of trace gases were introduced into a tree branch enclosure and a paired blank reference enclosure. The trace gas concentrations before and after the enclosures were measured, as well as the enclosed leaf area (single-sided) and gas flow rate to obtain the trace gas fluxes with respect to leaf surface. There was no detectable NO uptake for all tree types. The foliar NO₂ and O₃ uptake largely followed a diurnal cycle, correlating with that of the leaf stomatal conductance. NO₂ and O₃ fluxes were driven by their concentration gradient from ambient to leaf internal space. The NO₂ loss rate at the leaf surface, equivalently, the foliar NO₂ deposition velocity toward the leaf surface, ranged 0–3.6 mm s⁻¹ for bigtooth aspen, and 0–0.76 mm s⁻¹ for red oak, both of which are ~90% of the expected values based on the stomatal conductance of water. The deposition velocities for red maple and white pine ranged 0.3–1.6 mm s⁻¹ and 0.01–1.1 mm s⁻¹, respectively, and were lower than predicted from the stomatal conductance, implying a mesophyll resistance to the uptake. Additionally, for white pine, the extrapolated velocity at zero stomatal conductance was 0.4 ± 0.08 mm s⁻¹, indicating a non-stomatal uptake pathway. The NO₂ compensation point was ≤60 ppt for all four tree species and indistinguishable from zero at the 95% confidence level. This agrees with recent reports for several European and California tree species but contradicts some earlier experimental results where the compensation points were found to be on the order of 1 ppb or higher. Given that the sampled tree types represent 80-90% of the total leaf area at this site, these results negate the previously hypothesized important role of a leaf-scale NO₂ compensation point. Consequently, to reconcile these findings, further detailed comparisons between the observed and the simulated in- and above-canopy NOₓ concentrations, and the leaf- and canopy-scale NOₓ fluxes, using the multi-layer canopy exchange model with consideration of the leaf-scale NOₓ deposition velocities as well as stomatal conductances reported here, are recommended.

# 1 Introduction

The reactive nitrogen species nitric oxide (NO) and nitrogen dioxide ($NO_2$) are key components in tropospheric oxidation chemistry, affecting air quality by triggering the production of ground-level ozone, secondary organic aerosol, and acid rain.

Forests cover 27% of the world's land surface, and 34% of the land area of the United States (FAO, 2016), and are an important land cover type in the continental cycling of NO and $NO_2$ (collectively termed $NO_x$). In remote and relatively unpolluted forests, the main source of NO is biogenic emission from soil microbial nitrification and denitrification processes. Once it escapes the soil, NO is transported through the canopy by turbulent mixing that is coupled to the atmosphere above the forest. During this time, NO participates in chemical reactions with trace species present in ambient air, primarily with ozone to form

$NO_2$. This happens on a relatively short time scale of tens to a few hundred seconds. During daytime, additional reactions may further transform $NO_2$ to other oxidized nitrogen species, but on a longer time scale (Min et al., 2014). Physical loss pathways of $NO_x$ within the canopy include dry deposition and leaf stomatal and cuticular uptake. The relative differences in the time scales of the turbulent mixing and the chemical and physical sink processes determine the amount of $NO_x$ removed within the canopy, with the remaining $NO_x$ being released into the boundary layer.

The effect of leaf stomatal and cuticular uptake on the release of soil-emitted $NO_x$ through forest canopy to the atmosphere is described using an empirical parameter, the canopy reduction factor (CRF), introduced by Yienger and Levy (1995) for application in large-scale atmospheric chemistry studies that generally rely on the so-called "big-leaf" approach to represent atmosphere-biosphere exchange without considering the inhomogeneity of the loss processes within the canopy. Based on a parameterization using leaf area index and stomatal area index, it was estimated that 25−55% of soil-emitted $NO_x$ is lost within

forest canopies annually or seasonally depending on forest type. Those estimates of the effective release of soil $NO_x$ were further corroborated in a study by Ganzeveld et al. (2002) using, instead of the big-leaf approach, a multi-layer canopy exchange model in a chemistry-climate model. Additionally, by including the influences of wind speed, turbulence, and canopy structure when calculating the CRF, Wang et al. (1998) estimated that up to 70% of $NO_x$ was removed within the canopy in Amazon in April, agreeing with earlier results (Jacob and Wofsy, 1990). Accounting for both forests and other types of

ecosystems, Wang et al. (1998) also estimated the global average canopy reduction at 20%, versus 50% by Yienger and Levy (1995). More recently, Delaria et al. (2018) investigated $NO_x$ exchange with the leaves of *Quercus agrifolia* (California live oak) and obtained deposition velocities of $NO_2$ and NO under light and dark conditions. Implementing these results in a multi-layer single-column model, it was calculated that California oak woodland canopy removes 15-30% of soil-emitted $NO_x$, and other forests in California and Michigan, close to 60% (Delaria and Cohen, 2020).

Similarly, vegetation and plant surfaces also affect ozone levels through dry deposition (Clifton et al., 2019, 2020; Silva and Heald, 2018; Kavassalis and Murphy, 2017). In forested areas, ozone dry deposition occurs through leaf stomata as well as non-stomatal pathways including cuticular uptake, and wet or dry leaf surface reactions, while some $O_3$ is also removed by gas-phase chemical reactions e.g. with biogenic volatile organic compounds (BVOCs) and NO. Though these processes have been identified, the exact partitioning between the dry deposition pathways (and in-canopy chemical destruction) has not been

unequivocally determined, hindering the ability to correctly assess ground-level ozone. Thus, forest canopy plays a significant role in regulating the trace gas compositions in the atmosphere. Direct observations of $NO_x$ exchange with a wide variety of plants and in various ecosystems are necessary to achieve a better understanding of their overall impacts.

There have been over a dozen field and laboratory studies aimed at understanding leaf-level $NO_x$ uptake conducted since the 1990s, but primarily on European tree species (Raivonen et al., 2009, and references therein). From direct measurements of
foliar $NO_x$ uptake, a reasonably detailed understanding of the gas exchange processes between $NO_x$ and $O_3$ and plant leaves has been developed. Plants absorb $NO_2$ and $O_3$ mainly through leaf stomata, but also by leaf cuticular uptake (Chaparro-Suarez et al., 2011; Geßler et al., 2002; Coe, 1995; Rondón et al., 1993). The uptake efficiency varies across plants and is influenced by environmental conditions. Studies at leaf level and within leaves have found that after entering the stomata, $NO_2$ is metabolized through dissolution and enzyme-catalyzed reactions (Hu et al., 2014; Vallano and Sparks, 2008; Weber et al.,
1998; Nussbaum et al., 1993). Unlike $NO_2$ and $O_3$, foliar exchange of NO is small to insignificant (Hereid and Monson, 2001; Rondón et al., 1993), except for herbicide-treated soybeans (Klepper, 1979) and nutrient-fed sugar cane, sunflower, corn, spinach, and tobacco plants (Wildt et al., 1997), where NO emission was observed. Results from (Delaria et al., 2018) are consistent with these earlier findings.

In addition to $NO_x$ and $O_3$ deposition fluxes, $NO_2$ compensation points have also been obtained by extrapolating the linear
relationship between $NO_2$ flux and the ambient $NO_2$ concentration over the leaf surface (Raivonen et al., 2009; Slovik et al., 1996). The compensation point is the specific $NO_2$ ambient mole fraction or concentration at which $NO_2$ uptake by the plant leaves or $NO_2$ flux toward the leaf surface becomes zero. Reported values for this $NO_2$ compensation point ranged from 0.3 to over 3 ppb, depending on tree type and the conditions under which the measurements were made. The existence of such a point implies that when ambient $NO_2$ is below these thresholds, for example, in remote, unpolluted forest areas where it is usually
less than 1 ppb, the soil-emitted $NO_x$ would not be efficiently removed by the forest canopy necessary for balancing the $NO_x$ budget in the overlaying atmosphere above the forest. In fact, for those relatively clean conditions, the forest foliar would provide an additional atmospheric $NO_x$ source.

This conundrum, discussed by Lerdau et al. (2000), seemed to be resolved in the past decade when additional leaf-scale experiments were made using a new chemiluminescent $NO_x$ detector equipped with a highly $NO_2$ specific blue light converter
(Breuninger et al., 2012, 2013; Chaparro-Suarez et al., 2011). The improved $NO_2$ detection specificity of the instrument prevented artifacts caused by augmentation of the $NO_2$ signal from other nitrogen compounds such as nitrous acid (HONO), nitric acid ($HNO_3$), and peroxyacyl nitrates (PANs). These artifacts may have caused an observed reduction of $NO_x$ uptake that led to the conclusion of an (inferred) compensation point. The above work, on several native European trees, showed either a lower $NO_2$ compensation point than previously measured, at 0.05 to 0.65 ppb, or values not significantly different
from zero at the 95% confidence interval, and do not support the possibility of a foliar $NO_x$ source. However, when analyzing the observed $NO_x$ and $O_3$ concentrations in a North American hardwood forest at the University of Michigan Biological Station (UMBS) research site using a multi-layer canopy exchange model, Seok et al. (2013) found that the best agreement between simulated and measured $NO_x$ concentrations was obtained when a 1 ppb $NO_2$ compensation point was invoked. Further analysis

to assess the sensitivity of the simulated $NO_x$ mixing ratios to the representation of soil NO emission, leaf surface photolysis of nitrate, or advection was not able to reproduce the observations especially regarding the diurnal cycle of $NO_x$.

In order to verify these findings regarding the potential role of an $NO_2$ compensation point for the UMBS site, we conducted further field experiments on leaf-level gas exchange in summer 2016. This work is the first direct observation of foliar gas exchange of $NO_x$ and $O_3$ on mature trees growing naturally in a North American forest. To our knowledge, there has been one early study on young seedlings of several American tree species (Hanson et al., 1989), and one recent study on seedlings of California live oak (Delaria et al., 2018). In this work, we used a branch enclosure technique to measure mainly $NO_2$, as well as NO, and $O_3$ exchange rates at the leaf surface of four locally dominant tree species, *Pinus strobus* (eastern white pine), *Acer rubrum* (red maple), *Populus grandidentata* (bigtooth aspen), and *Quercus rubra* (red oak). Results obtained from these measurements provide information to reassess the possibility of a foliar $NO_2$ source and the role of the canopy on $NO_x$ and $O_3$ cycling at this forest site. In this paper, we use both "uptake" and "foliar deposition" when describing trace gas exchange at the foliar level. Both terms refer to the process of trace gas loss upon contact with the leaf surface; but generally, the subject of "uptake" is the plant whereas the subject of "deposition" is the trace gas.

## 2 Experiment

### 2.1 Site description

The experiments were carried out at the Program for Research on Oxidants: PHotochemistry, Emissions, and Transport (PROPHET) research site at UMBS, which occupies about 10,000 acres on the northern tip of the Lower Michigan Peninsula (45.56° N, 84.71° W, Fig. 1). The area was heavily logged until the end of the 19th century. It also experienced several severe wildfires from 1880 to 1920. Natural reforestation started when the location was acquired for the research station in 1909. Today, bigtooth aspen, trembling aspen (*Populus tremuloides*), red maple, red oak, and white pine dominate within about a 1-km radius of the PROPHET site, whereas within a 60-m radius of the site, there are more white pine trees and almost no trembling aspen.

The northern part of the peninsula is fairly remote. The air is free from anthropogenic pollutants unless meteorological conditions result in the advection of air masses from surrounding major cities: to the southwest, Chicago, IL and Milwaukee, WI; to the southeast, Detroit, MI; and to the east, Toronto, ON. During the field experiments, about 35% of the time, air masses were coming from these directions. However, since the enclosures were purged with scrubbed ambient air (see Methods section), the direct influence of pollutants on the enclosed plant material was minimal.

The enclosure measurements were carried out from July 20 to August 3, 2016. Sky conditions were sunny to mixed sun and clouds most of the time. The average ambient temperature measured in the canopy (~2 m from the ground) was 24°C during daylight, and 18°C at night, with maximum and minimum temperatures of 31°C and 12°C, respectively. There were two main rainfall events, with the most recent being two days prior to the start of the enclosure experiments. The average soil temperature

was near 19°C throughout the experiment period, and the soil moisture decreased gradually after the rainfall. These conditions are within the normal ranges for this site in July.

## 2.2 Methods

Branch enclosure experiments were conducted sequentially on branches of white pine, red maple, bigtooth aspen, and red oak. The estimated ages of the white pine, red maple, and red oak trees were about 15 to 20 years, and the bigtooth aspen, 5−10

years. All tree branches were selected based on their sun exposure, accessibility, and size. The height of the enclosed branches ranged from 3 to 10 m above the ground.

     The enclosure system was composed of three parts: the enclosures, the airflow system, and the trace gas measurement instruments (Fig. 2). The enclosure, essentially a flow chamber, was constructed using a 61 x 91 cm bag made of Tedlar® (polyvinyl fluoride) (Jensen Inert Products, Florida, USA) with three factory-installed 0.95 cm diameter ports to attach tubing

and sensor wires. The branch was carefully enclosed by the bag so that it was situated as close to the middle of the bag as possible. The open end of the bag was then closed around and tied onto the main stem of the branch, tight enough to secure the enclosure when it was inflated by the purge air, but also with enough leakage to allow air to escape during purging. Each branch enclosure was paired with an identical enclosure assembly without any plant material as the background reference to account for wall effects and other factors that may affect trace gas concentrations. The reference enclosure was placed adjacent

to the branch enclosure but without obstructing the sunlight to the enclosed tree leaves.

     Air delivery and air sample lines (polytetrafluoroethylene or PTFE), each about 30 m long, were connected to the enclosures and to the instruments housed in an air-conditioned trailer at the site. Between the trailer and the enclosures, the air and sample lines were bundled together and sheathed inside black flexible insulation hoses linked together end to end. The hoses were wrapped in aluminum foil to keep the sample lines from absorbing heat from sunlight.

Ambient air from outside the trailer and scrubbed free of dust, $O_3$, and $NO_x$ was used as the purge gas. An oil-free air compressor (Medo USA, now Nitto Kohki USA) was used to pull the ambient air through an organic vapor/acid gas respirator cartridge (Magid, Illinois, USA), which functioned as a dust filter. Downstream of the compressor, the air was further filtered by an ozone scrubber (Thermo Fisher Scientific), activated charcoal, and a $NO_x$ scrubber (Purafil, Inc., Georgia, USA). After the filters, polytetrafluoroethylene (PTFE) tubing (12.7 mm OD and 9.52 mm OD) was used to carry the air to the enclosure

chambers. The tubing was connected to the port on the Tedlar bag at the end near the tip of the enclosed branch, opposite the bag opening. Inside the bag connected to the same port was an air distributor made from a loop of tubing (9.52 mm OD) with pinholes (~1.5 mm diameter) about 1 cm apart drilled along its entire length. This allowed even distribution and mixing of the purge air insider the enclosure. The flow rate of the purge air was maintained at 37 L m$^{-1}$. The volume of the inflated enclosure was ~57 L, giving the air a residence time of ~1.5 min. Additional resident time of sample air due to the sample line (30 m,

3.175 mm ID) was ~ 6 sec.

     For the trace gas exchange experiments, known amounts of $NO_2$, NO, or $O_3$ were added into the scrubbed air stream. $NO_2$ and NO were from compressed standard gas cylinders (Scott-Marrin, Inc., California, USA), and $O_3$ was made in situ using a

mercury Pen-Ray lamp $O_3$ generator (UVP, California, USA) and compressed zero air. A KOFLO® type mixer was placed just downstream of the trace gas inlet to ensure even mixing of the added trace component with the scrubbed air.

NO, $NO_2$, $O_3$, $CO_2$, and $H_2O$ mixing ratios before and after the enclosures were measured, with the air sample selected using a set of solenoid valves. The concentrations of these gases were calculated using the ideal gas law and the measured air temperature. The time for each sample was five minutes, alternating between the enclosure inlet and outlet. The reference enclosure inlet and outlet were sampled once an hour. The environmental conditions were also recorded, including ambient and enclosure temperatures (S-THB, Onset Computer Corp., Massachusetts, USA), leaf temperatures (thermocouple wire

sensors, Omega Engineering, Connecticut, USA), relative humidity (S-THB, Onset), leaf wetness (for qualitative assessment of leaf conditions only) (S-LWA, Onset), and photosynthetically active radiation (PAR) (S-LIA, Onset). Standard commercially available instruments were used for $O_3$ (Model 49i, Thermo Fisher Scientific, USA) and $CO_2/H_2O$ (LiCor 840, Li-Cor Corp., Nebraska, USA). NO and $NO_2$ were measured using a home-built chemiluminescence detector that utilizes the light-emitting reaction of NO with $O_3$ (Ryerson et al., 2000).

The $NO_x$ instrument was programmed to run on 5-minute cycles, each with a one-minute measurement of zero air (UHP, Airgas, USA), followed by a 2-min measurement of NO, and a 2-min measurement of $NO_2$. In $NO_2$ mode, $NO_2$ was first converted to NO, then measured the same as in the NO mode. The conversion was done using an LED UV light source (L11921-500, Hamamatsu Photonics). The peak light emission of this LED was at $385 \pm 5$ nm, matching the absorption peak of $NO_2$ and minimizing the interference from the unwanted photolysis of HONO. The $NO_2$ to NO conversion efficiency was

~ 0.68. Because the ambient air was scrubbed to remove $O_3$ (and other trace gases) before entering the enclosures, the effect of ambient $O_3$ on $NO_x$ measurements was negligible. A high-concentration (1.5 ppm) NO standard dynamically diluted with ultra-high purity zero air (Airgas, USA) was used to calibrate the NO measurement. For the $NO_2$ calibration, NO in the same diluted standard was partially converted to $NO_2$ by adding a controlled amount of ozone (generated in situ using a Pen-Ray ozone generator and 99.98% oxygen). The instrument calibration runs were initiated automatically about every 7 hours during

regular operation. The overall $1\sigma$-precision for a 5-minute measurement cycle was ~2 ppt for NO and ~4 ppt for $NO_2$. The accuracy of the NO and $NO_2$ measurements was ~30 ppt.

Water vapor at the enclosure inlet and outlet was measured using a LiCor 840. The instrument was calibrated using a LiCor dew point generator. The ambient relative humidity results from the Onset sensors were compared with the data from a nearby AmeriFlux tower (within 100 m) (Vogel, 2016a), and the agreement was within 3%. It was noticed that on particularly hot and

humid afternoons, there was condensation of water in the sample line leading to the instruments. The condensed water was removed promptly with gentle warming of the affected section of the sample line. The data recorded during these times were excluded.

After installation, each set of branch and reference enclosures was first purged with scrubbed ambient air at least overnight and through the early morning hours (up to ~10:00 local time) to allow the branches to acclimate, and also to reduce the amount

of any possible surface-deposited photochemically labile compounds that might interfere with the measurements (Raivonen et al., 2006). The gas exchange experiments were then started and carried out for the following 2 to 4 days. A known amount of

the trace gas was introduced into the purge air flow. This included zero concentration, i.e. purging with scrubbed air between the trace gas additions. The maximum mixing ratios of the trace gases in the purge air were kept within the range of typically observed ambient measurements, i.e. $NO_2$ <1.2 ppb, NO<300 ppt, and $O_3$<60 ppb.

The enclosed leaves were harvested after the completion of the measurements and immediately placed in an oven to be air-dried at 65°C. The leaf area was then measured by forming a monolayer of the dried leaves on graph paper. The enclosed single-sided leaf area for white pine, red maple, bigtooth aspen, and red oak was 0.35, 0.26, 0.11, and 0.44 $m^2$, respectively.

Leaf-level uptake or emission of the trace gas leads to a trace gas concentration difference between the enclosure inlet and outlet. In the enclosure, the flux of the trace gas with respect to the leaf surface is:

$$F_x = \frac{Q}{A}(c_o - c_i) \, , \hspace{10em} (1)$$

where $F_x$ (pmol m$^{-2}$ s$^{-1}$) is the flux of the trace gas $x$; $c_i$ and $c_o$, in pmol m$^{-3}$, are the trace gas concentration measured at the enclosure inlet and outlet, respectively. $Q$ (m$^3$ s$^{-1}$) is the purge air flow rate. $A$ (m$^2$) is the one-sided area of the enclosed leaves, as the stomata, the part of leaf anatomy most relevant to gas exchange, generally are located on the underside of tree leaves (Kirkham, 2014). A resulting flux with a negative sign reflects the loss of the trace gas at the leaf surface, and a positive flux,

emission from the foliage. All the trace gas concentration changes through the branch enclosure $(c_o - c_i)$ were corrected against the background obtained from the reference enclosure before the fluxes were determined according to Eq. (1). The detection limit of flux, i.e. the minimum absolute value above which the flux is significantly non-zero ($p<0.05$, or at the 95% confidence level), was determined using the flux data obtained during the scrubbed air purge at nighttime when no emission from the leaves was expected because generally the leaf stomata are closed at night. These detection limits (Table 1) reflect

the measurement precision of the instruments, variations of the actual enclosure conditions over time, and fluctuation of the purge air flow rate. Nighttime transpiration in trees and shrubs has been measured in prior work, with reports of nighttime transpiration rates ranging from 0 to as much as 25% of the daytime value (Dawson et al., 2007), suggesting that leaf stomata may remain open at night for some plants. However, this possibility did not affect the above results as there was no evidence of a consistent concentration difference above zero between the enclosure outlet and inlet measurements.

**2.3 Tree branch samples**

The representative tree species were determined based on the basal and leaf area coverage within the 60 m radius of the research site. Tree branches for the enclosure experiments were selected for their accessibility from the ground. Preferences were given to those with adequate sun exposure and to mature trees whenever possible. Enclosed branches of white pine (*Pinus strobus*) and red maple (*Acer rubrum*) were ~7 m above ground and from trees that were over 10 m in height. The enclosed branches

of red oak (*Quercus rubra*, ~6 m) and bigtooth aspen (*Populus grandidentata*, ~4.5 m) were ~3 m above ground.

## 3. Results

We first examine the results obtained when only scrubbed air (without any addition of $NO_x$ or $O_3$) flowed through the enclosures. The NO and $NO_2$ fluxes during the scrubbed air purge are shown in Fig. 3 along with the PAR and leaf temperature measured at the same time. Each data point represents a 5-minute measurement. The data points in gray are indistinguishable from zero within the 95% confidence interval based on the detection limits listed in Table 1. Those outside this confidence interval are marked by black symbols. It is expected that after the plant enclosures are conditioned with the hours-long scrubbed air purge, there will be no signal of $NO_x$ at the enclosure outlet unless there is a source within the enclosure to supply a detectable amount of $NO_x$. Indeed, for most of the day, there was no detectable amount of $NO_x$ (or $O_3$) at the enclosure outlet. However, besides a few scattered data points that are outside the confidence interval, there also appear to be some consistent positive fluxes of $NO_2$ lasting around 30 min or less occurring around noon or early afternoon. Because of the relatively short duration of this $NO_2$ emission during the brightest time of the day, it is unclear whether the flux was due to emission from leaves, or due to the photolysis of any oxidized nitrogen substrate remaining on the leaf surface even after the initial overnight and morning purging. The mixing ratios corresponding to the observed fluxes were less than 30 ppt in each of the enclosures. Below we present the results from each of the trace gas addition experiments.

### 3.1 $NO_2$

Nitrogen dioxide ($NO_2$) was introduced into the purge air at different concentrations between zero to 40 nmol m$^{-3}$ (~1 ppb). Generally, when $NO_2$ was added to the enclosure, there was a negative flux, indicating uptake of the trace gas by the plant material (Fig. 4, top panel). The magnitude of the flux was proportional to the input $NO_2$ concentration. In addition, when the input concentration was held constant for several hours or overnight, the flux had a diurnal pattern, e.g. bigtooth aspen on July 30 to July 31. It was lowest at night, increased through the morning hours and peaked around midday before diminishing again toward nighttime. This behavior strongly suggests that the $NO_2$ uptake by these trees is in large part controlled by leaf stomatal aperture and, at the same time, driven by the $NO_2$ concentration gradient from the air around the leaf surface to the leaf internal space.

There are a couple of factors that complicated the $NO_2$ gas exchange experiment. First, the $NO_2$ standard used for delivering $NO_2$ to the enclosure contains about 5% NO that was unavoidably added to the enclosure. Secondly, when there was intense direct sunlight, some $NO_2$ in the enclosure was photolyzed. While corrections for these interferences were done using the measurements from the reference enclosure, it is difficult to completely remove the artifact caused by $NO_2$ photolysis. This is because the sunlight exposure of the two enclosures, although situated side-by-side, was often uneven, and the measurements of the enclosures were done not simultaneously but sequentially. This problem is particularly pronounced for clear sky conditions with strong contrasts in sunlit and shaded conditions inside the canopy. The branch enclosure was always positioned to get more sun exposure than the reference enclosure if choices needed to be made. Therefore, the branch enclosure likely received more sunlight overall, even though it might be more shaded during some measurement cycles. Generally, for the

periods of strong sunlight, there is residual NO after the correction against the reference enclosure is made. If we assume all this is due to an underestimation of $NO_2$ photolysis and make a further correction by combining the changes of NO and $NO_2$,

the data quality is not improved while more noise is introduced to the data. Because of this and because we are not absolutely certain about all possible sources of $NO_x$ from the branch enclosures, we prefer to adhere with the correction using only the reference enclosure measurements and view the resulting $NO_2$ flux as an upper bound, with possibly as much as 20% overestimation under direct sunlight conditions, which accounts for ~16% of all data during the $NO_2$ exchange experiments. The size of the stomatal aperture, regulated by the plant's need to optimize photosynthesis and simultaneously minimize water

loss, can be gauged by stomatal conductance of water using Eq. (2) (Weber and Rennenberg, 1996):

$$g_{H_2O} = F_{H_2O} / (C_{H_2O\_leaf} - C_{H_2O\_enclosure}), \qquad (2)$$

in which the flux of water ($F_{H_2O}$, in mmol m$^{-2}$ s$^{-1}$) due to plant transpiration is calculated by applying the measured water concentration difference at the inlet and outlet of the enclosure to Eq. (1). $C_{H_2O\_leaf}$ (mmol m$^{-3}$), the water concentration inside the leaf air space, is calculated using the measured temperature of the enclosed leaves, assuming the air in the leaf internal

space is saturated with water vapor. $C_{H_2O\_enclosure}$ (mmol m$^{-3}$), the water concentration of the branch enclosure, is evaluated using the measured enclosure relative humidity and temperature data. The resulting stomatal conductance of water, $g_{H_2O}$ (mm s$^{-1}$), for the four enclosed branches is shown in Fig. 5a. The stomatal conductance has a clear diurnal pattern, mainly following the daily cycles of sunlight and photosynthesis. The magnitude varies from tree to tree. The conductance of the white pine and the red maple branches were similar, ranging from near zero at night to about 3 mm s$^{-1}$, while the conductance of red oak was

0 to ~1 mm s$^{-1}$, and the bigtooth aspen, 0 to ~6 mm s$^{-1}$. When the conditions are such that the difference between the leaf and air temperatures is small and the enclosure humidity is high, the difference between $C_{H_2O\_leaf}$ and $C_{H_2O_{enclosure}}$ is also reduced, increasing the uncertainty in $g_{H_2O}$. In our measurements, this happened mostly from dawn to sunrise, accounting for ~10% of the total data points, where the ($C_{H_2O\_leaf} - C_{H_2O\_enclosure}$) was within one standard deviation from zero.

Knowing the stomatal conductance of water, the expected rate of $NO_2$ deposition through the plant stomata can be calculated.

Across the stomata, the deposition is a diffusion-controlled process (Weber et al., 1998; Weber and Rennenberg, 1996), where the expected rate is the product of the stomatal conductance of water multiplied by the square root of the ratio of the molecular weight of water to the molecular weight of $NO_2$:

$$g_x = g_{H_2O} \times \sqrt{\frac{MW_{H_2O}}{MW_x}}, \qquad (3)$$

where $g_x$ (mm s$^{-1}$) represents the expected stomatal uptake rate for the trace gas species $x$ (here $x = NO_2$); $g_{H_2O}$ (mm s$^{-1}$) is

the stomatal conductance of water; and $MW$ (g mol$^{-1}$) represents molecular weight.

From the measurements, the leaf-level $NO_2$ deposition velocity, $v_{dNO_2}$ (mm s$^{-1}$), is the $NO_2$ flux toward leaf surface ($F_{NO_2}$, in pmol m$^{-2}$ s$^{-1}$) normalized to the corresponding $NO_2$ concentration in the enclosure ($c_{o,NO_2}$, in pmol m$^{-3}$):

$$v_{dNO_2} = F_{NO_2} / c_{o,NO_2}. \qquad (4)$$

If $NO_2$ deposition is exclusively controlled by stomatal uptake, agreement between the measured deposition velocity and the calculated stomatal uptake rate is expected, i.e. $v_{dNO_2} = g_{NO_2}$. If not, additional factors, such as internal mesophyll resistance (Gut, 2002; Thoene et al., 1996), or leaf cuticular adsorption (Geßler et al., 2002; Coe, 1995; Rondón et al., 1993), may also play a role with the former decreases and the latter increases the overall foliar deposition velocity.

In Fig. 5b, the measured foliar deposition velocity of $NO_2$ is plotted together with the calculated stomatal uptake rate for comparison. The agreement is generally good for all experiments, suggesting that the foliar deposition of $NO_2$ for these tree species is indeed closely related to stomatal aperture. This also suggests that the effects of internal mesophyll resistance and cuticular uptake of $NO_2$ are relatively minor. The strength of correlation between $NO_2$ deposition and stomatal conductance is evaluated using the Pearson correlation coefficient, $\rho$, which has a possible value between -1 to 1, with a value of 0 indicating no correlation and a value that is away from zero indicating increasing positive or negative correlation. In Fig. 6, the foliar $NO_2$ deposition velocity is plotted against the stomatal conductance for each tree. The correlation coefficient for bigtooth aspen is 0.96 and for red oak is 0.85, both showing a strong positive correlation between the deposition velocity and stomatal conductance. The correlation is also evident but relatively weaker for the white pine ($\rho = 0.73$) and the red maple ($\rho = 0.71$).

The relationship between the deposition velocity and stomatal conductance is also examined using linear regression analysis. If $NO_2$ deposition is entirely controlled by stomata, the deposition rate at zero conductance ($g_{H_2O} = 0$), when the stomata are closed, should be zero; and the slope of the deposition rate vs. stomatal conductance should be equal to $\sqrt{\frac{MW_{H_2O}}{MW_{NO_2}}}$, or 0.62 (recall Eq. (2)). This relationship is shown in Fig. 6 with the solid blue line. The best fit and the 95% confidence bounds are represented by the red solid and dashed lines. Also listed in the figure are the slope (m), the intercept (b), and the $r^2$ value of each fit. The linear relationship for bigtooth aspen appears to be the tightest, with over 90% of the data variation can be explained by the fit. The intercept is nearly zero, and the slope of 0.56 is close to 0.62, making it reasonable to conclude that for the bigtooth aspen, stomatal uptake dominates $NO_2$ loss at the leaf surface. A similar conclusion can be made for red oak, where $r^2$ is 0.72, the slope and the intercept are 0.54 and 0.03, respectively.

The red maple is different. The slope of $NO_2$ deposition rate to stomatal conductance is 0.25, far less than 0.62. The data also appear to have more scatter. On the time series plot (Fig. 5a), the stomatal conductance on the morning of July 25 (from 8:30 to 12:30) shows high variability that is not reflected by the $NO_2$ deposition rate at the same time. Possibly an unknown measurement issue for water concentration during this time or sources of water exchange at leaf surface other than stomata (see Discussion section 4.1) led to the high variability. However, excluding this portion of the observations and using only the data obtained prior to this time window, from 13:00 on July 24 to 8:00 on July 25, resulted in a modestly improved linear fit with a slope still below 0.3. For white pine, the slope of $v_{dNO_2}$ vs. $g_{H_2O}$ is 0.40, also lower than the expected value of 0.62 based on stomatal-controlled diffusion. These lower than expected slopes imply there may exist mesophyll resistance to $NO_2$ uptake for these tree species. Such resistance to stomatal uptake of $NO_2$ has previously been observed on some trees such as European *Picea abies* (Norway spruce) seedlings (Thoene et al., 1996), and Amazonian *Laetia corymbulosa* and *Pouteria glomerate* (Gut, 2002). However, in a separate study of Norway spruce seedlings (Rondón and Granat, 1994), no evidence of

internal resistance to $NO_2$ stomatal uptake was found. Researches on $CO_2$ diffusion and $H_2O$ transport into leaf internal spaces have revealed that mesophyll resistance is subject to environmental perturbations, and the responses among and within species can vary (Xiao and Zhu, 2017). It is reasonable to assume that the mesophyll resistance to $NO_2$ uptake may also subject to
environmental conditions, and systematic observations under different conditions are needed to obtain more general conclusions.

The y-intercept of the fitted line accounts for any possible additional foliar deposition when the stomata are closed and consequently the stomatal conductance is zero. Of all four trees studied, only white pine has an intercept significantly larger than zero at $0.43 \pm 0.09$ mm $s^{-1}$, indicating a possible role of wet leaf surfaces and/or cuticular uptake. See Discussion section
4.1 below. The nighttime stomatal conductance of white pine is relatively high, with a median value at 0.57 mm $s^{-1}$, compared to 0.05-0.19 mm $s^{-1}$ for the other trees, probably due to incomplete stomatal closure at night (Dawson et al., 2007). There is corresponding nighttime deposition of $NO_2$, with a higher rate for white pine relative to the other trees (Fig. 5).

### 3.1.1 Compensation point of $NO_2$

To determine at what concentration the $NO_2$ flux becomes zero, we plot the $NO_2$ flux vs the enclosure $NO_2$ concentration in
Fig. 7. The amplitude of the stomatal conductance for each data point is represented by the color scale, with cool to warm colors corresponding to stomatal conductance from low to high in each enclosure. As expected, the flux increases with increasing $NO_2$ concentration in the air surrounding the leaves; and at a given concentration, the flux increases with stomatal conductance. For each enclosure, we selected the data points taken when the stomatal conductance was at least 50−60% of its maximum measured during the experiments, indicated by the large, warm-colored symbols in Fig. 7. These data were then fit
with linear regression for flux vs $NO_2$ concentration. The intercepts of the best-fit regression line and the zero-flux line, representing the compensation point, are listed in Table 2. For all four tree types within the range of stomatal conductance considered, the inferred $NO_2$ compensation point is well below 100 ppt, and not distinguishable from zero within measurement uncertainties.

### 3.2 NO

Nitric oxide (NO) was added to the purge air at concentrations up to ~10 nmol $m^{-3}$ (~250 ppt) to the white pine, red maple, and bigtooth aspen enclosures (Fig. 4b). Red oak was not included in this experiment. No significant NO flux toward the leaf surface was observed. This agrees with observations made on Scots pine (Rondón et al., 1993), corn leaves (Hereid and Monson, 2001), and *Quercus agrifolia* (Delaria et al., 2018). On the contrary, for the white pine, a positive flux up to 2.7 pmol $m^{-2}$ $s^{-1}$ from the enclosure was measured when NO was added (Fig. 4b, White Pine), indicating emission from the enclosed
plant material. This flux also appears to increase with the enclosure NO mixing ratio. Although the photolysis of surface deposited nitrogen oxides may cause such positive NO flux, during the scrubbed air purge prior to the addition of NO, there was no significant NO emission from the pine enclosure. That said, this experiment was done only once in a span of five hours

from late morning to early afternoon. We cannot absolutely rule out possible interference from nitrogen containing chemical components in the system.

### 3.3 $O_3$

Up to 2.2 μmol m$^{-3}$ (~55 ppb) of ozone ($O_3$) was introduced to the enclosures. As in the case of $NO_2$, there was an $O_3$ loss within the enclosure, and it increased with input $O_3$ concentration (Fig. 4c). Shown in Fig. 5c is the comparison of the measured foliar $O_3$ deposition velocity to the expected stomatal uptake rate calculated using leaf stomatal conductance and the square root of the ratio of the $O_3$ and $H_2O$ molecular weight, $\sqrt{\frac{MW_{H_2O}}{MW_{O_3}}}$. For red maple, bigtooth aspen, and red oak, these two values agree reasonably well, implying that foliar ozone loss is mainly through leaf stomatal and closely related to stomatal conductance. Correlation analyses were not performed here due to the limited number of data points.

For white pine, the measured leaf-level $O_3$ deposition velocity is significantly greater than the expected stomatal uptake rate by a factor of 2 or more. It is known that on average up to 60% of ozone deposition in vegetated areas is through non-stomatal pathways (Clifton et al., 2019). Within a branch enclosure, non-stomatal pathways can include deposition to wet leaf surfaces (Zhou et al., 2017; Altimir et al., 2004), cuticular uptake, chemical reactions at the leaf surface (Jud et al., 2016; Fares et al., 2010) and in the gas phase with biogenic organic compounds (BVOCs). Estimation of the possible contribution from gas-phase reactions with BVOCs was made as follows. The upper bounds of typical emission rates at 30 °C and PAR level at 1000 μmol m$^{-2}$ s$^{-1}$ for monoterpenes and other BVOCs (excluding isoprene) are 3 and 5 μg C g$^{-1}$ h$^{-1}$, respectively (Guenther et al., 1994). The speciation of major BVOCs emitted by white pine at UMBS is based on Kim et al. (2011), including α- and β-pinene, limonene, linalool, α-humulene, and β-caryophyllene. Using the rate constants of the BVOCs with ozone reactions (Burkholder et al., 2015), and the residence time of 1.5 min in the enclosure plus ~6 sec in the sample line before reaching the detector, the estimated ozone loss due to gas-phase chemical reactions was less than 1%. Even with optimal light and temperature conditions for BVOC emission, the estimated gas-phase chemical removal would only be on the order of a few percent.

## 4 Discussion

### 4.1 Foliar trace gas exchange: $NO_2$ and $O_3$

For two weeks during the summer PROPHET2016 campaign, we examined the leaf-level $NO_2$, NO, and $O_3$ gas exchange of four different tree species. This work provided a first insight into the general characteristics of the gas exchange of these North American trees in their natural habitat. The trees used in the enclosure measurement represent 80% of the total leaf area within a 1000 m radius of the research site, and 90% within the 60 m radius. It is evident from the results that bidirectional foliar gas exchange depends on the trace gas in question and tree type, and is influenced by diverse and complex environmental conditions, similar to the findings from previous studies mainly on European tree species and on annual plants (grasses and

crops). The foliar uptake rates of $NO_2$ and $O_3$ vary from tree to tree and even within the same tree. Leaf stomatal conductance of $H_2O$ emerges as a strong indicator of the uptake efficiency. The foliar $NO_2$ deposition of bigtooth aspen and red oak is

almost entirely controlled by stomatal aperture. For red maple and white pine, the correlation coefficient is over 0.7, even though the measured $NO_2$ foliar deposition velocity is 40–50% of the predicted stomatal uptake rate. Except for white pine, the $O_3$ foliar deposition velocities of all the studied trees also covary with stomatal conductance (Fig. 5c). Generally, the leaf-level $NO_2$ and $O_3$ deposition velocity can largely be inferred from the stomatal conductance of water only, as also concluded from earlier studies of European tree species (Breuninger et al., 2013; Rondón and Granat, 1994).

Thus, the factors controlling leaf stomatal conductance would in turn greatly influence $NO_2$ and $O_3$ deposition in a forested environment. These factors include PAR level, ambient temperature, moisture, and soil conditions, as well as ambient $CO_2$ (Jarvis, 1976). Further, the capability of the foliar uptake of trace gases would also depend on the intrinsic characteristics of leaf stomata, such as their size and density on the leaf surface, determined by plant species and stage of maturity, and factors such as growth history, leaf age, tree height and the vertical location of the leaf on the tree (Kirkham, 2014; Sparks et al., 2001;

Schäfer et al., 2000). In this work, the stomatal conductance of the bigtooth aspen was 3–5 times higher than that of the other trees. Biological features, such as plant and leaf age, stomatal density, may have contributed to this difference. Compared with the other three trees in this work, the aspen was younger and smaller. The enclosed branch was in the upper part of the crown containing developing new leaves. Past measurements, albeit on different species, have shown that for the same species under similar environmental conditions, leaves of young trees generally have higher stomatal conductance than old ones (Niinemets,

2002; Hubbard et al., 1999; Fredericksen et al., 1995; Yoder et al., 1994). Another possible reason for the observed high $g_{H_2O}$, while direct evidence has yet to be found, is the number of stomata. Many trees have stomata on only the lower (abaxial) leaf surface; however, trees that belong to the genus Populus, which includes aspen, are an exception. They have stomata on both sides (amphistomatous), a feature that allows increased photosynthetic rate and fast growth (Kirkham, 2014). If the bigtooth aspen leaves are indeed amphistomatous, a relatively high $g_{H_2O}$ can be expected. We compared the environmental conditions

of the enclosures. The integrated PAR exposure levels were similar. The daily variation of the relative humidity in the bigtooth aspen enclosure was not significantly different from the others. In contrast, the average daily temperature was 19.2 °C, cooler than the temperatures (23.9 °C, 22.6 °C, and 21.6 °C) in the other enclosures, similar to the average ambient air temperature outside the enclosure during the same time, 19.1 °C, and 23.6 °C, 22.4 °C, and 21.3 °C. The combined conditions of moisture and temperature led to a relatively low vapor pressure deficit (VPD) in the aspen enclosure, 0.8 kPa, compared to 1.2 kPa

(white pine), 1.0 kPa (red maple), and 1.4 kPa (red oak) in the others. Generally, VPD and $g_{H_2O}$ are inversely correlated and a low VPD corresponds to a relatively high $g_{H_2O}$ (Urban et al., 2017a, 2017b; Hubbard et al., 1999). However, because here we are comparing different tree species, we consider the observed results to stem from the combination of biological and environmental factors. Further examination of these factors is beyond the scope of this paper, nevertheless, it would be beneficial to take this temporal and spatial variability and inhomogeneity into account in model parameterizations of trace gas

dynamics since plant stomata are the main conduit of $NO_x$ and $O_3$ deposition over vegetation.

When extrapolated to zero stomatal conductance, the deposition velocity of $NO_2$ to white pine was 0.43 mm s$^{-1}$ (Figure 6a), implying deposition unrelated to leaf stomata, possibly to wet leaf surfaces and/or to leaf cuticula. This observation does not exclude the possible existence of these pathways when the stomata are open. A deposition velocity higher than expected based on the stomatal conductance would result if there is significant non-stomatal deposition. On the other hand, mesophyll resistance renders a lower deposition velocity than the expected value. There is no mechanistic reason why the deposition velocity associated with either a non-stomatal pathway or mesophyll resistance should remain constant or vary linearly with stomatal conductance. The relationship of deposition velocity, $v_{dNO_2}$, and stomatal conductance, $g_{H_2O}$, would remain essentially linear as long as stomatal deposition dominates or the non-stomatal deposition term is constant while mesophyll resistance is small. However, if mesophyll resistance is significant, it would limit the increase of $v_{dNO_2}$ with stomatal conductance.

To assess the role of wet leaf surfaces to non-stomatal deposition, we calculated the white pine enclosure dew point using the temperature and relative humidity data and compared it to the measured leaf temperature. The leaf temperature was always higher than the dew point during the experiments, excluding the possibility of a wet leaf surface from the condensation of pure water. However, a  microscopic water film may nevertheless form at a relative humidity as low as 50% if there are hygroscopic deposits on the leaf surface (Sun et al., 2016; Burkhardt and Hunsche, 2013; Burkhardt and Eiden, 1994). The microscopic water film could potentially modify gas exchange rates of water-soluble trace gases in the air. Data from this work do not contain information that can be used to delineate the possibilities of trace gas dissolution into microscopic water films or cuticular uptake. Further investigations with appropriately designed experiments, better measurement precisions, longer observation time, and under different environmental conditions are necessary to delineate the various possible deposition pathways and their dependencies.

To put our results in perspective, we compare our measured daytime maximum foliar deposition velocity of $NO_2$ with the results from previous studies on American trees (Table 3). Although the development stages of the trees, PAR, humidity and temperature conditions are different, the results are comparable, ranging from 0.76 to 1.6 mm s$^{-1}$ from this work and 0.4–1.8 mm s$^{-1}$ from earlier work. We also compare our results to those of several native European trees - scots pine, evergreen oak, common oak, European beech, and silver birch, measured under the conditions of PAR = 900 µmol m$^{-2}$ s$^{-1}$, maximum temperature 27.7°C, and relative humidity 31.2–99.9% (Breuninger et al., 2013). The maximum $NO_2$ deposition rates were ~0.5–1 mm s$^{-1}$ for all but the birch tree, which was ~1.5 mm s$^{-1}$. These numbers are also fairly similar to those of pine, maple and oak reported here. What stands out but without a direct or closely related comparison is the high rate of trace gas uptake by the aspen leaves. Although the comparisons show reasonable agreement, it is evident that the $NO_2$ (and $O_3$) foliar uptake is highly variable depending on a myriad of conditions, both environmental and intrinsic to tree species and developmental stage. Measurement results and comparisons from different studies are probably also sensitive to experimental protocols and environmental conditions. These factors should be taken into consideration if more comparisons are to be made in future work.

### 4.1.1 NO$_2$ compensation point

Measured fluxes of NO$_2$ toward the leaf surface while the stomatal conductance was at least 50% of the observed maximum value were used to assess the possible existence of an NO$_2$ compensation point. It would have been indicated by a zero or positive NO$_2$ flux at significantly non-zero NO$_2$ concentrations defined by the measurement system detection limit (Table 1). We found no such evidence of an NO$_2$ compensation point for all the tree species measured in this work. Indeed, this lack of evidence of a compensation point is also supported by the fact that no significant, sustained NO$_2$ emission was observed while the enclosures were purged with the scrubbed air only. For all four trees in this study, the compensation point, if it exists at all, would be well below 150 ppt. Thus, this finding does not support the existence of a 1 ppb NO$_2$ compensation point as suggested in the previously mentioned combined NO$_x$ concentration measurement and canopy exchange model study (Seok et al., 2013) to reach the best agreement between the simulated and observed NO$_x$ concentrations above and within the forest canopy at the UMBS site. We would like to point out that the NO$_2$ flux may approach zero even at high NO$_2$ concentrations if the stomata are not adequately open and the stomatal conductance is lower than the values used above (Fig. 7). However, because here the NO$_2$ uptake is mainly through stomata, such zero flux at relatively high NO$_2$ mixing ratios is not indicative of a compensation point; rather, it is from the reduced capacity of absorbing NO$_2$ under reduced stomatal conductance. Our result agrees with recent reports on several other tree species that an NO$_2$ compensation point is not observed above the detection limit of the measurement using improved NO$_2$-specific instruments with minimal interference from other nitrogen compounds (Breuninger et al., 2013; Chaparro-Suarez et al., 2011).

### 4.2 NO

There was no significant leaf-level deposition of NO for all the tree species studied here. Instead, relatively small NO emissions were detected from white pine when up to ~250 ppt NO was added to the enclosure. Delaria et al (2018) reached the same conclusion from their study on *Quercus agrifolia*. We searched for possible errors that might have led to the results but could not find an obvious explanation. Certainly, additional measurements are necessary to verify this observation. Using the leaf area index of white pine at UMBS, 0.11 m$^2$/m$^2$ (Vogel, 2016b) and the maximum measured flux, 2.7 pmol m$^{-2}$ s$^{-1}$, we estimated the potential canopy-wide NO flux from this emission to be 0.3 pmol m$^{-2}$ s$^{-1}$, less than 10% of the reported minimum soil NO emission flux of 4–10 pmol m$^{-2}$ s$^{-1}$ at UMBS (Nave et al., 2011).

Although this observation seems counterintuitive, in previous publications, emission of NO has been reported from leaves of individual corn plants exposed to 0.1–0.3 ppb NO (Hereid and Monson, 2001), from leaves of California live oak exposed to air containing NO (Delaria et al., 2018), from several nitrate-nourished plant species (Wildt et al., 1997) as well as pesticide-treated soybean leaves (Klepper, 1979). Additionally, recent plant physiological studies have started to reveal the mechanism of plant NO production and its importance for regulating growth and development, immunity, and signaling (Astier et al., 2017; Río, 2015; Yu et al., 2014), as well as for responding to pollutants and stress (Bison et al., 2018; Farnese et al., 2017;

Velikova et al., 2008). In light of these advances, more targeted observations of foliar NO exchange probably should be conducted while taking these biological factors into consideration.

## 5 Summary and Conclusions

Using a branch enclosure technique and with controlled addition of trace gases, we obtained data on NO, $NO_2$, and $O_3$ leaf-level gas exchange from field experiments on several native tree species in a northern hardwood forest in Michigan, USA. To our knowledge, this is the first time such experiments have been done on North American tree species in a field study. The results provided a new dataset of $NO_x$ and $O_3$ leaf-scale fluxes and have allowed comparisons of the gas exchange characteristics of mature trees compared to seedlings of these species in the lab and to mature European tree species in the field. The data also provide information, including an upper bound on $NO_2$ compensation points for these trees, to models of $NO_x$ and $O_3$ dynamics at the canopy level, particularly for the forest at the PROPHET research site.

A brief survey of the foliar $O_3$ loss found that uptake by the deciduous trees also closely followed stomatal conductance, while the $O_3$ foliar deposition velocity for white pine was much larger than expected from leaf stomatal uptake alone. Removal via gas-phase chemical reactions was calculated to be negligible based on estimates of known BVOC emission rates and speciation, implying other non-stomatal pathways - cuticular uptake, dissolution to wet leaf surfaces, and/or chemical reactions at the leaf surface - are responsible for the additional ozone deposition, with their relative importance to be determined.

The trace gas exchange characteristics of NO, $NO_2$, and $O_3$ at the leaf level varied depending on tree type and environmental conditions. For NO, there was no measurable foliar uptake from any of the trees studied here. On the contrary, there appeared to be a small emission of NO from white pine when NO was added to the enclosure. Leaf-level $NO_2$ uptake of bigtooth aspen and red oak was mainly through leaf stomata, with the leaf-level deposition velocity of $NO_2$ closely following predicted values based on the stomatal conductance of water and molecular diffusivity. The stomatal conductance of aspen was ~5 times higher than that of red oak (and thus the foliar $NO_2$ deposition velocity for aspen was also much higher). Because stomatal conductance is subject to a variety of factors including those intrinsic to plants, further investigation is needed to determine whether this difference is generally associated with the plant species or is environmentally driven. For white pine and red maple, the foliar $NO_2$ deposition velocity correlated with stomatal conductance, but there were additional factors that prevented deposition from increasing as much as expected with increasing conductance, suggesting the existence of internal mesophyll resistance to uptake. Furthermore, for white pine, there was foliar $NO_2$ deposition when stomatal conductance was zero, suggesting a non-stomatal $NO_2$ loss pathway such as cuticular uptake.

The possible existence of an $NO_2$ compensation point was inferred by examining the linear relationship between $NO_2$ flux and ambient $NO_2$ concentration when the stomata were open, and the stomatal conductance was at least 60% of the maximum measured value. The results showed that the compensation point was $\leq 60$ ppt for all trees and was statistically indistinguishable from zero within the measurement sensitivity. This finding does not support the suggested 1 ppb

compensation point needed to reconcile the observed and model-simulated $NO_x$ mixing ratios by Seok et al. (2013). Neither does it support any significant foliar $NO_2$ emission from these tree species at low ambient $NO_2$ conditions.

It is noteworthy that, beyond the findings in Seok et al. (2013), inclusion of an $NO_2$ compensation point not only provided the best agreement in terms of $NO_x$ concentrations, but also gave the best agreement between simulated and observed atmosphere-biosphere $NO_x$ fluxes at UMBS in summer 2016 (J. Murphy, personal communications, 2018). Evaluations of

these simulations with the Multi-Layer Canopy CHemistry Exchange Model (MLC-CHEM), which was used in Seok (2013) have not yet included a direct comparison with the leaf-scale $NO_x$ and $O_3$ fluxes reported here. Such a comparison could address both the observed large differences in the magnitude of the stomatal conductance for specific trees and its diurnal cycle, focusing on the early morning onset of stomatal opening and uptake. This would further confirm whether there is a leaf-scale $NO_x$ emission flux due to an $NO_2$ compensation point, or if a strongly reduced $NO_x$ uptake might partially explain the

observed dynamics in the above- and in-canopy $NO_x$ concentrations. This analysis would also benefit from more detailed temporally and vertically resolved $NO_x$ concentration gradient observations compared to the Seok et al. (2013) study, which we measured in conjunction with the leaf-level work described here. This comparison is an essential next step in attempting to reconcile the findings of this study with previous studies of $NO_x$ exchange at the UMBS forest, and will be presented in a follow-up publication.

Our findings confirmed that the main conduit of trace gas foliar uptake is leaf stomata. A thorough grasp of the trace gas uptake efficiency hinges on an understanding of the leaf stomatal properties, which depend on the genetic makeup and developmental stage of the plant, as well as the environmental conditions of sunlight, water vapor, ambient temperature, soil, and nutrients. Meanwhile, the additional factors affecting foliar trace gas exchange, such as mesophyll resistance, cuticular uptake, and stress responses, are also subject to plant intrinsic and external conditions and remain to be better understood.


*Acknowledgments.* This research was funded by the National Science Foundation (NSF Grant AGS-1561755). Three anonymous reviewers gave thoughtful comments and questions that helped to improve the manuscript. We thank Professor S. Bertman of Western Michigan University for providing training and access to equipment that made it possible to reach canopy-level tree branches for the enclosure work. We also thank Dr. T. Ryerson of NOAA Chemical Science Division for advice on

$NO_x$ measurement instrumentation, and Dr. C. Vogel of the University of Michigan Biological Station (UMBS) for help with housing and calibrations of our instruments during the campaign. Last but not least, we appreciate the support and help from all PROPHET-AMOS 2016 participants and the staff at UMBS.

*Data availability*. The data from this work are archived at https://umich.box.com/v/PROPHETAMOS2016.


*Author contributions*. WW, SR, and JH constructed the enclosure chambers, deployed the instruments, and carried out the field experiments. WW performed the data analysis and prepared the manuscript. LG and DH provided extensive comments and

suggestions for the manuscript. LG and DH initiated this project based on previous fieldwork at UMBS and model analysis regarding leaf-level gas exchange.


*Competing interests*. The authors declare that they have no conflict of interest.

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

**Figures and Tables**

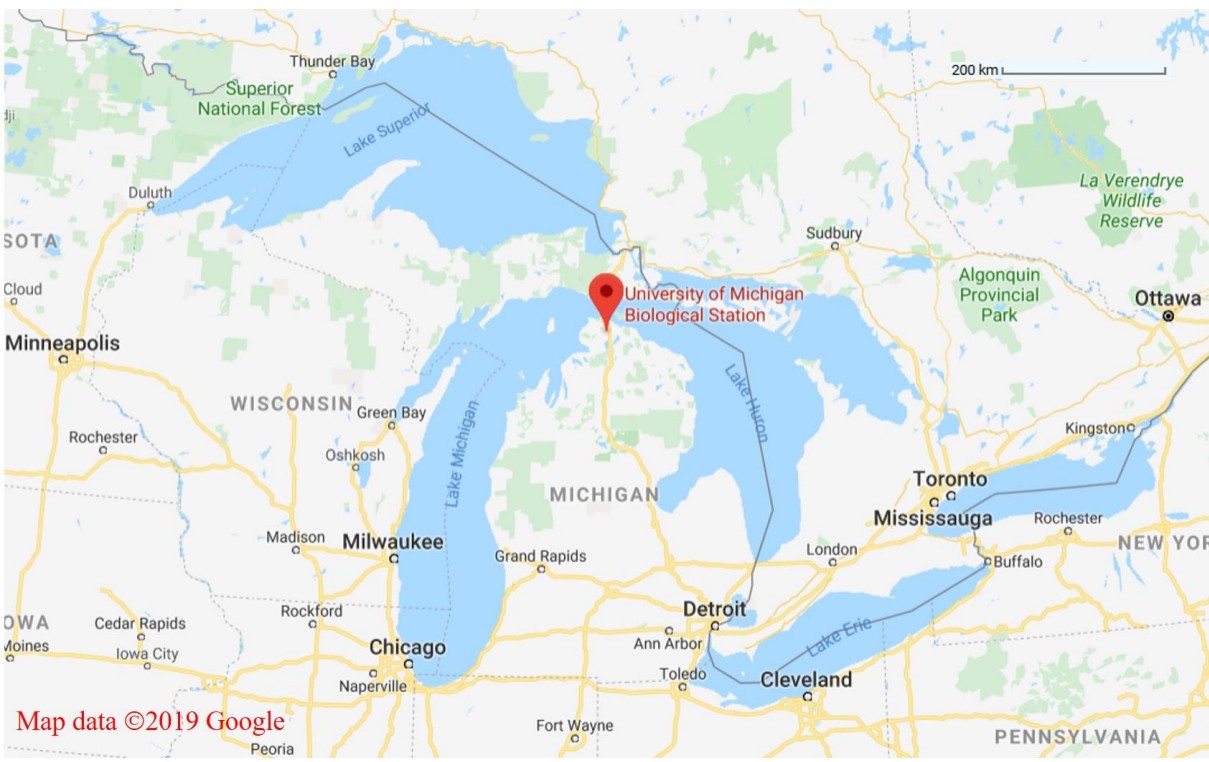


**Figure 1. Location of the University of Michigan Biological Station, (45.56°N, 84.71°W), indicated by the red pin on the map. The map scale is shown in the upper right corner. (Map data ©2019 Google).**

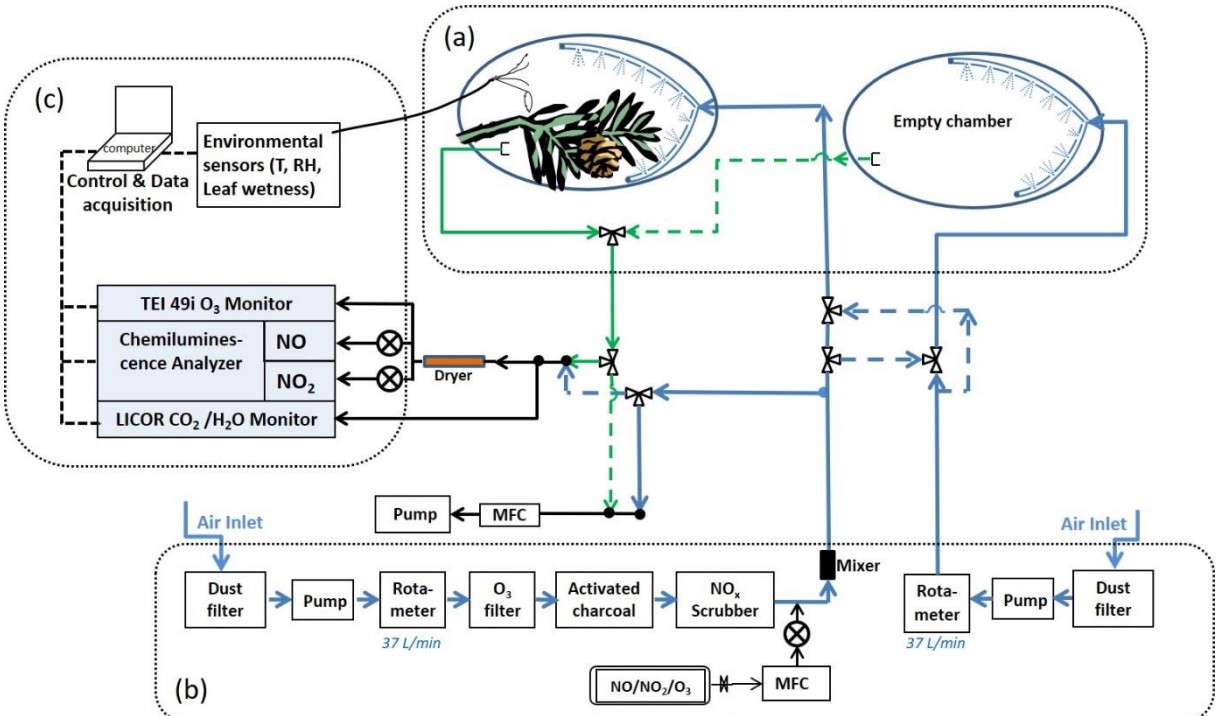

Figure 2. Schematic of the enclosure experiment system. The system is comprised of three main parts as shown in the figure: (a) the enclosures, (b) the purge air flow system, and (c) the trace gas measurement instruments. The blue lines and arrows indicate the air flowing into the enclosures; the green lines and arrows indicate the air flowing out of the enclosures; and the black lines and arrows indicate the air sample flow and the balance flow (to maintain constant flow rates in the enclosures). NO and NO$_2$ gas standards were used for the controlled addition of these trace gases to the input air stream. Controlled O$_3$ addition was done by generating ozone on demand using a pen-ray lamp and ultra-high purity oxygen. See text for instrument details.

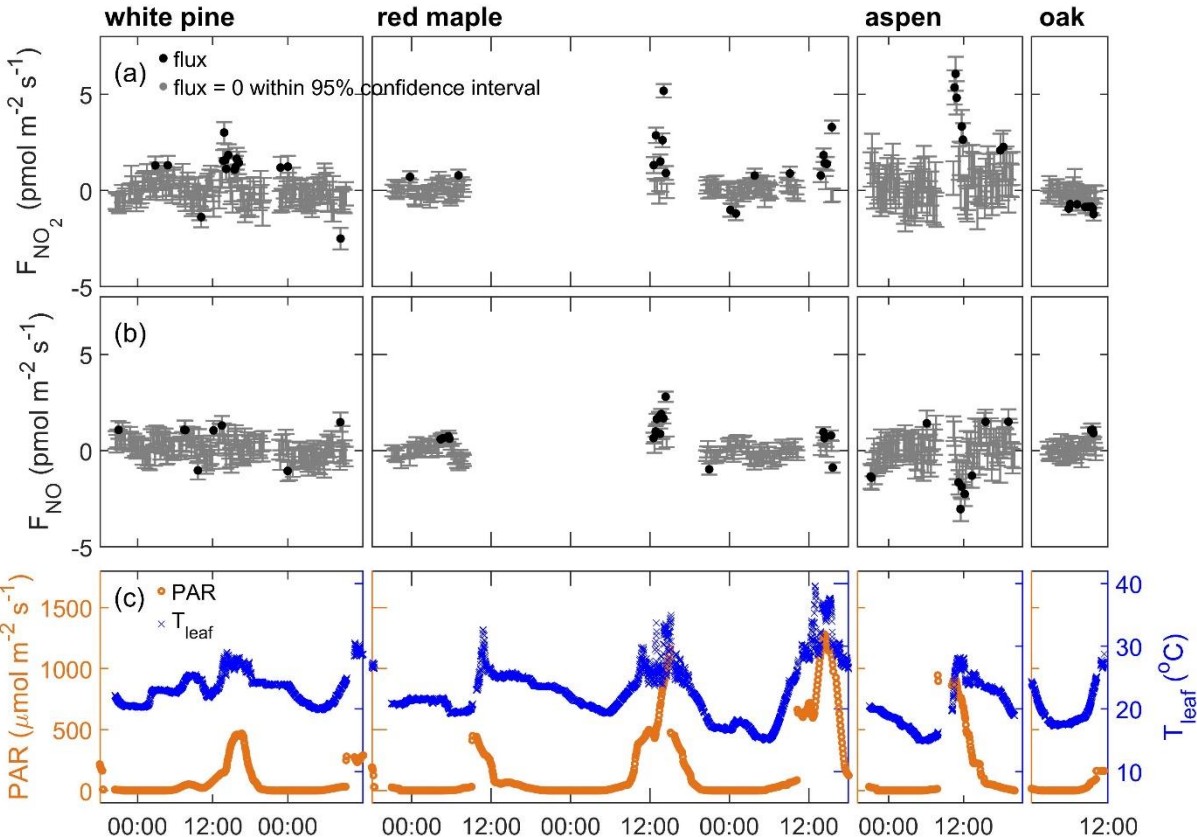

**Figure 3. Apparent fluxes of NO₂ and NO when the enclosures were purged with scrubbed air. From left to right each panel corresponds to the enclosure of white pine, red maple, bigtooth aspen, and red oak. From top to bottom: (a) NO₂ flux, (b) NO flux, and (c) PAR (left axis, orange) and temperature of the enclosure leaves (right axis, blue). In panel (a) and (b), fluxes that are indistinguishable from zero within the 95% confidence interval are represented by gray dots; statistically significant fluxes are represented by black dots; error bars represent 1-σ measurements uncertainties propagated through the calculations.**


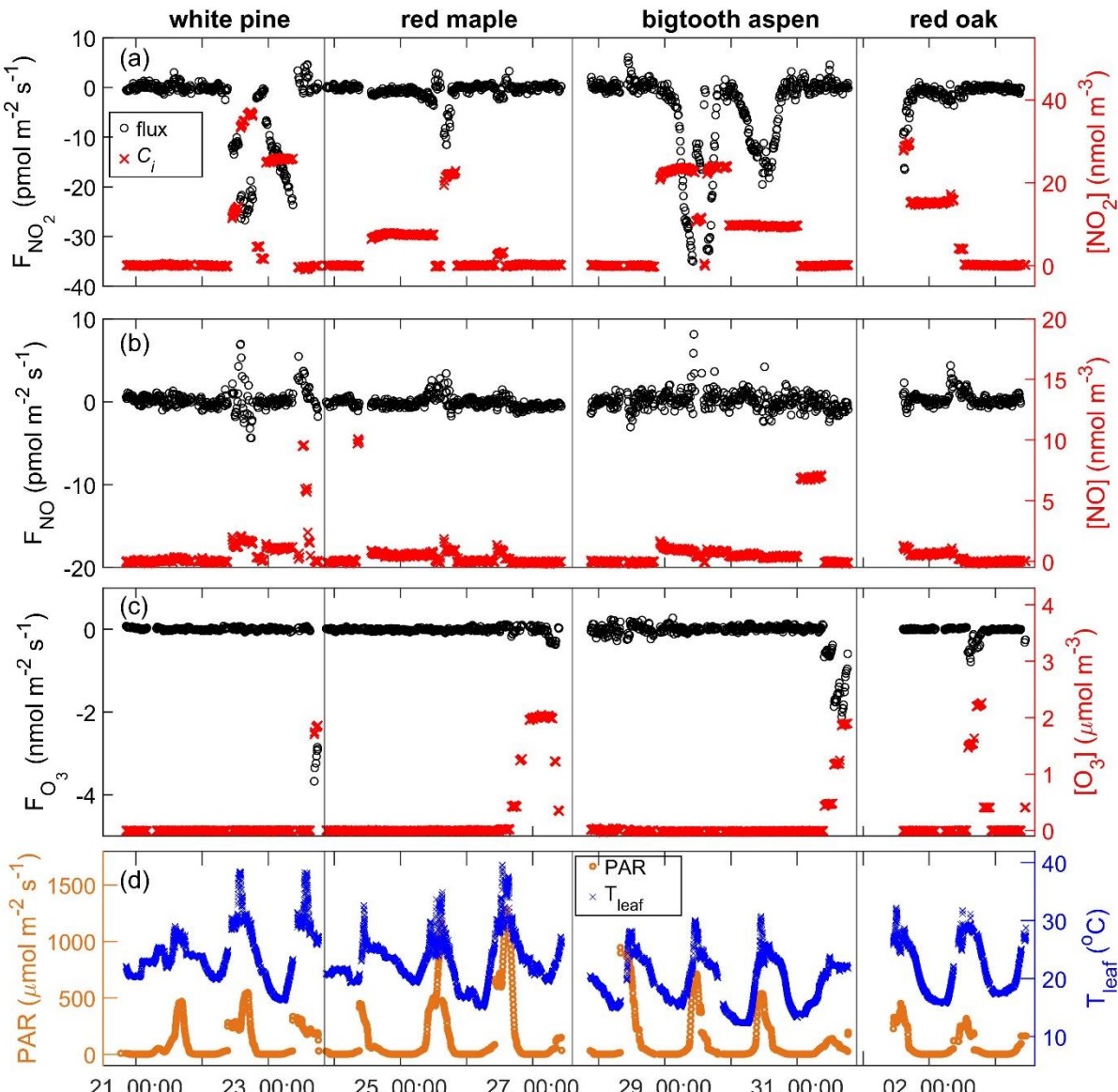

**Figure 4.** Time series of the enclosure gas exchange experiments from July 21 to August 3, showing the trace gas fluxes (black symbols: o, left axis) and input trace gas concentrations, $C_i$, (red symbols: x, right axis) of, (a) $NO_2$, (b) NO, and (c) $O_3$. Solar irradiation PAR (orange: o, left axis) and temperature of the enclosure leaves (blue x, right axis) are shown in the bottom panel (d). The tree species are labeled for each enclosure period at the top of the figure. The x-axis tick label format is Day, HH:MM.

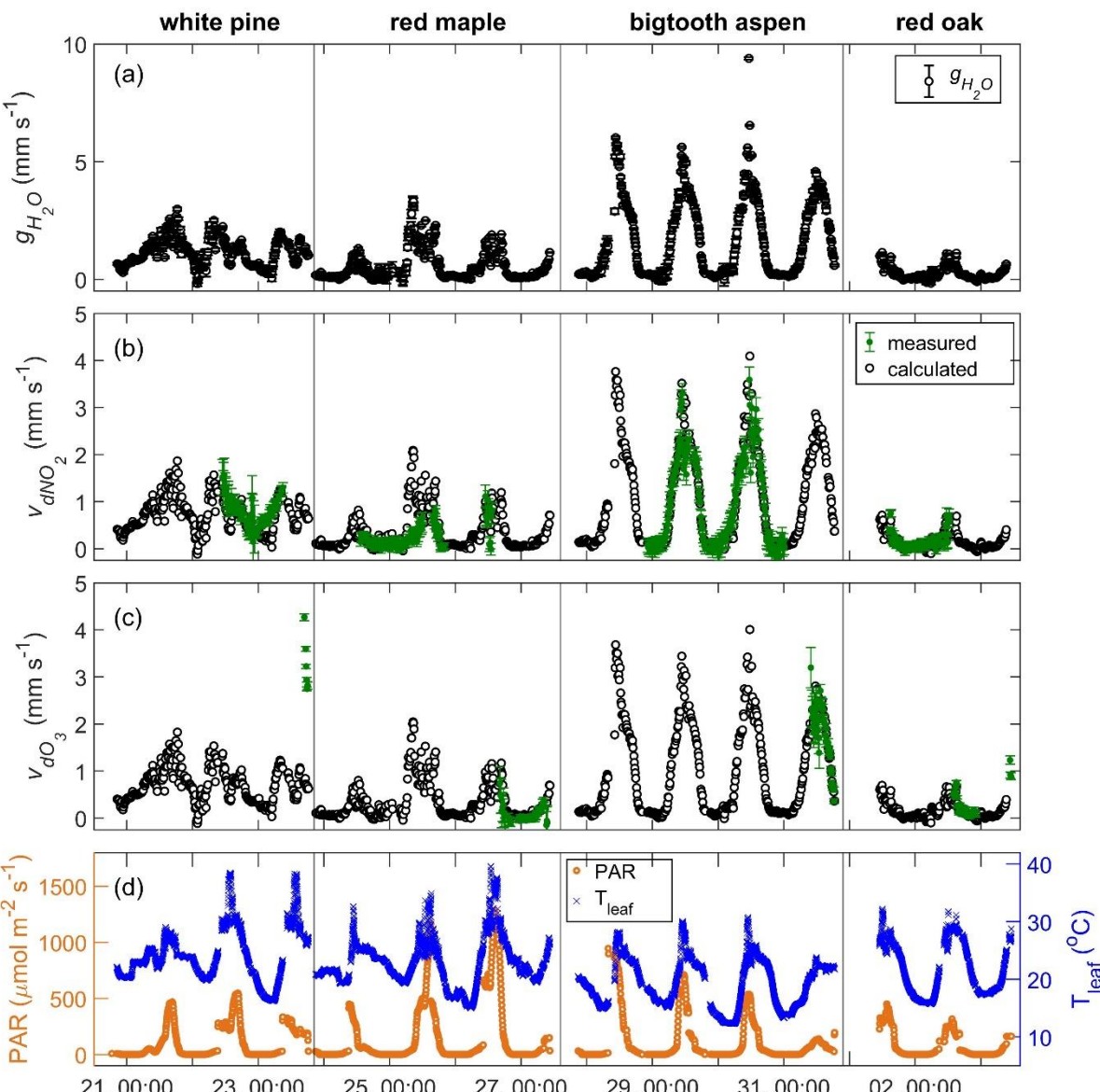

**Figure 5.** Time series plots of (a) the measured stomatal conductance of water; (b) the measured foliar deposition velocity (green symbols) and the calculated stomatal uptake rate (black symbols) of $NO_2$; (c) the measured foliar deposition velocity (green symbols) and the calculated stomatal uptake rate (black symbols) of $O_3$; and (d) the corresponding PAR and leaf temperature during the experiments from July 21 to August 3, 2016.


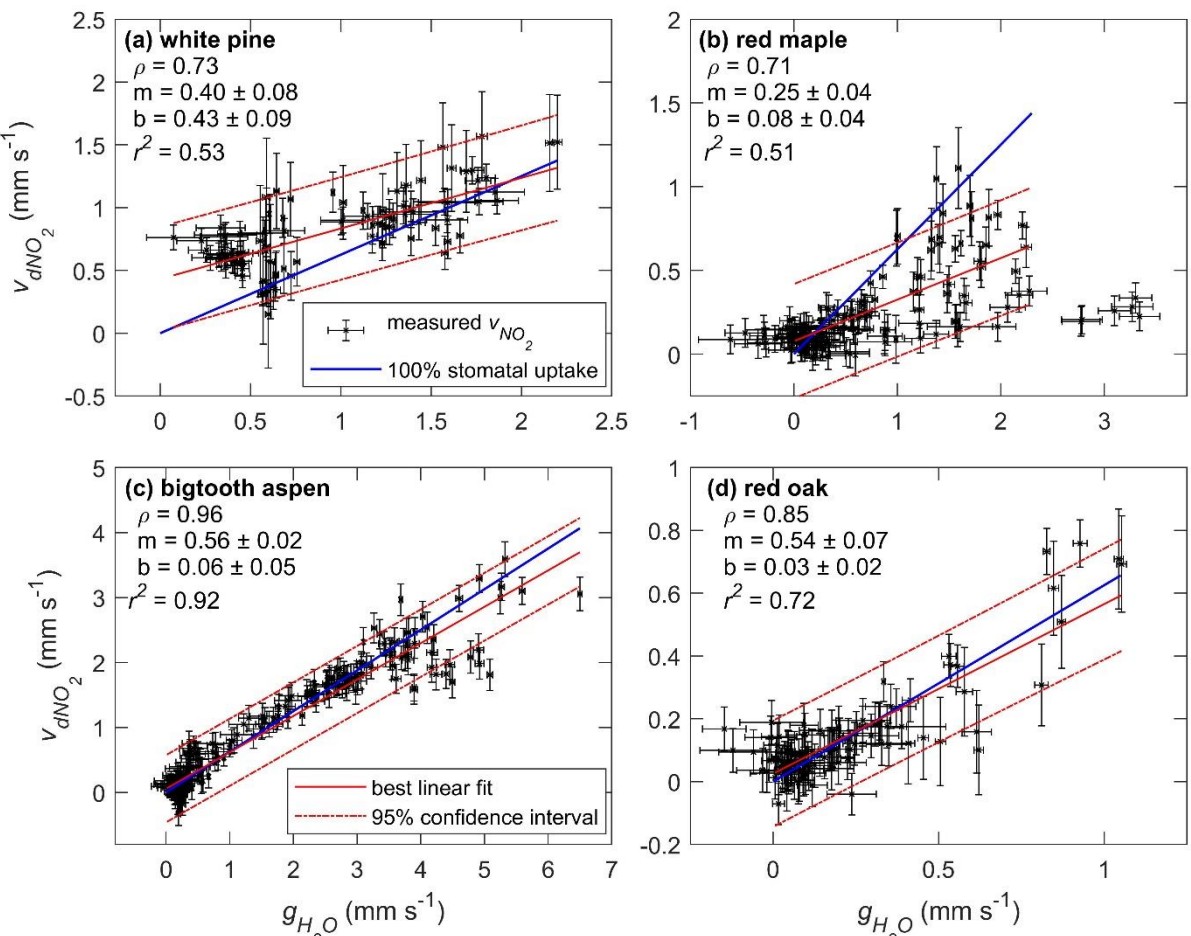

**Figure 6: Scatter plots of foliar NO₂ deposition velocity ($v_{dNO_2}$) vs. stomatal conductance of water ($g_{H_2O}$) for (a) white pine, (b) red maple, (c) bigtooth aspen, and (d) red oak. The data points and their error bars are represented by the black symbols. The solid and dashed red lines are the best-fit linear regression and the 95% confidence bounds, respectively. The solid blue line shows the relationship between the deposition velocity and the stomatal conductance if NO₂ loss is entirely controlled by the stomata. The slope of the blue line is 0.62, the square root of the ratio of the molecular weight of water to NO₂. Listed in each subplot under the tree name are the Pearson correlation coefficient (ρ), and the slope (m) and intercept (b) of the best fit linear regression line.**

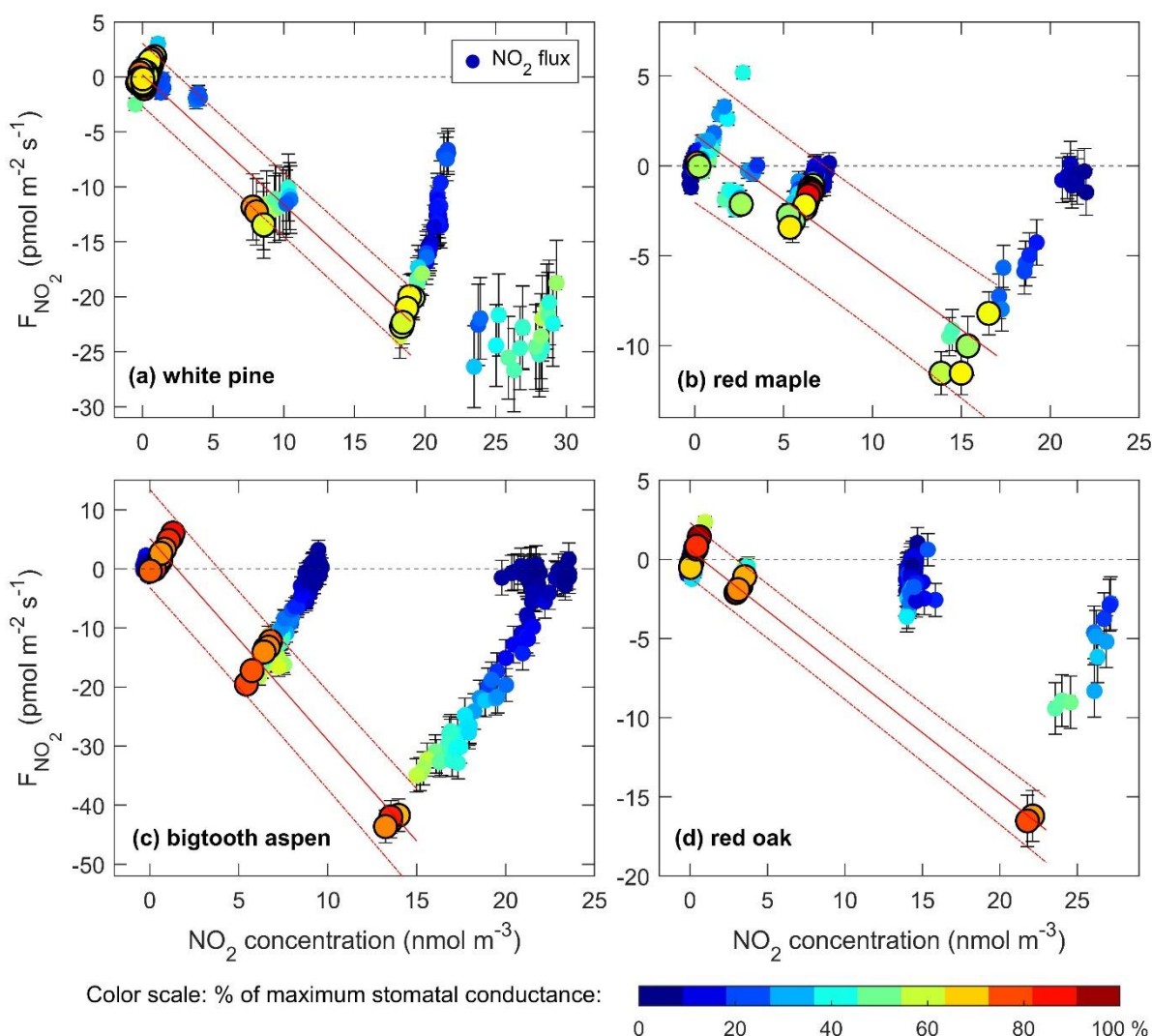

**Figure 7. NO₂ flux toward the leaf surface vs. NO₂ concentration in each enclosure. Also shown is the stomatal conductance on a cool–warm color scale with the dark blue representing the lowest values and the red the highest values observed. Flux data with the stomatal conductance 60% of the observed maximum or higher were used for the linear extrapolation to find the compensation point. These data points are shown as larger symbols with a black outline/border. The solid and dashed red lines show the linear fit and 95% confidence bounds. The resulting compensation points are listed in Table 2.**

**Table 1. Detection limits of the foliar fluxes of NO₂, NO, and O₃. These were determined using the flux data obtained during the nighttime scrubbed air purges when no foliar gas exchange was expected. These limits mainly reflect the measurement precision of the trace gas concentrations at the inlets and outlets of the branch and reference enclosures, variations of the enclosure conditions over time, and the fluctuation of the purge air flow rate.**

| Detection limit | $F_{NO_2}$ (pmol m$^{-2}$ s$^{-1}$) | $F_{NO}$ (pmol m$^{-2}$ s$^{-1}$) | $F_{O_3}$ (pmol m$^{-2}$ s$^{-1}$) |
|---|---|---|---|
| White pine | 1.1 | 1.0 | 76.3 |
| Red maple | 0.6 | 0.6 | 68.0 |
| Bigtooth aspen | 2.0 | 1.3 | 233 |
| Red oak | 0.8 | 0.8 | 42.7 |

**Table 2. Compensation points from the flux vs. concentration linear fits in Fig. 7. The ranges of the stomatal conductance of the data used are also listed.**

| Tree species | White pine | Red maple | Bigtooth aspen | Red oak |
|---|---|---|---|---|
| Stomatal conductance (mm s$^{-1}$) | 1.8–3.0 | 1.8–3.5 | 4.5–6.0 | 0.8–1.3 |
| Compensation point ± 95% confidence level (ppt) | 4 ± 60 | 60 ± 119 | 38 ± 59 | 19 ± 56 |
| Compensation point ± 95% confidence level (nmol m$^{-3}$) | 0.2 ± 2.4 | 2.4 ± 4.9 | 1.6 ± 2.4 | 0.8 ± 2.3 |

**Table 3: Comparison of foliar NO₂ deposition velocity from this work and earlier studies. The maximum velocity measured in each enclosure is listed with the corresponding light, RH, and leaf temperature at the time of the measurement.**

| Tree species | $v_{NO_2}$ (mm s$^{-1}$) | PAR (μmol m$^{-2}$ s$^{-1}$) | RH (%) | T_leaf (°C) | Source |
|---|---|---|---|---|---|
| *Pinus strobus* (white pine) | 1.6 | 601 | 67 | 30 | This work |
| *Pinus strobus* (white pine, seedling) | 0.4 | "Adequate to open leaf stomata" | n/a | 29.4 | Hansen (1989) |
| *Acer rubrum* (red maple) | 1.1 | 1200 | 72 | 30 | This work |
| *Acer rubrum* (red maple, seedling) | 1.8 | "Adequate to open leaf stomata" | n/a | 29.4 | Hansen (1989) |
| *Quercus rubra* (red oak) | 0.76 | 1086 | 61 | 28 | This work |
| *Quercus agrifolia* (California live oak) | 1.23 | 1190 | 50–65 | 26 | Delaria (2018) |
| *Populus grandidentata* (bigtooth aspen) | 3.6 | 850 | 71 | 25 | This work |