# Peer review of "Measurement report: Leaf-scale gas exchange of atmospheric reactive trace species (NO2, NO, O3) at a northern hardwood forest in Michigan"

_Atmospheric Chemistry and Physics, 2020_

## Referee Comment (RC1) · Anonymous Referee #1 · 14 Apr 2020

This a very thorough paper contributing to the body of literature on atmosphere-biosphere exchange of NOx and o3. It makes important steps to resolve inconsistencies between previous laboratory and field observation studies. It should be published after consideration of the minor comments below.

P2 L60: The citation of Delaria et al., 2018 and reported estimation of 15-30% removal of soil-emitted NOx is correct—oak woodlands have a very low LAI. However, in Delaria and Cohen 2020 (now published and not in discussion), they report much larger canopy reductions for forests with more typical LAI, in line with the 25-55% loss previously reported.

[Figure]

P3 L64: Extra parenthesis

P3 L83: Would be nice if the instrument were stated explicitly.

P3 L96: Correct "folia" to "foliar"

P7 L205: Several studies have observed significant stomatal opening during the night (e.g. Dawson et al., 2007–10.1093/treephys/27.4.561). Consider adding a discussion of how, if this was occurring in your chamber, this assumption would have affected your results (if at all).

P8 L246: A more detailed description of your empty chamber photolysis corrections would be useful.

P10 L314: Units for the intercept should be added. Additionally, under the resistance model framework you discuss, the relationship of Vd to gH2O is non-linear. How might this affect your inferences of cuticular uptake?

P12 L363: How high of emission rates would this require? It it outside the range reported for trees of the species considered?

P13 L395-396: should this be VPD?

P13 L403: "In the future,"? "In future work,"?

P14 L437: This was also a conclusion of Delaria et al., 2018.

O3 deposition: There are a number of recent references discussing ozone deposition that are not included. The paper would be stronger if it placed itself in the context of these and other recent papers on the subject (e.g. Silva and Heald GRL 2018 https://doi.org/10.1002/2017JD027278, Kavassalis and Murphy GRL 2017 https://doi.org/10.1002/2016GL071791 and Clifton et al. in Reviews of Geophysics 2020 https://doi.org/10.1029/2019RG000670).

―――――――――――――――――――――

---

## Referee Comment (RC2) · Anonymous Referee #2 · 20 Apr 2020

The study by Wang et al. investigates the existence of an NO2 compensation point for a mixed hardwood forest by using branch enclosures in the field. Foliar deposition of NO, NO2 and O3 were measured by applying different concentrations of these compounds in cleaned air to the enclosures to measure foliar uptake/emission. The enclosures were installed at mature trees of several dominant tree species. This study is very valuable as there are very few studies on NO2 deposition to trees and esp. to mature trees under field conditions. The topic is very important to understand NOx cycling and to finally come up with a more profound understanding of in-canopy NOx retention. This in turn is of utmost importance to adequately model NOx fluxes from forest canopies. The measurements and data analysis are sound and the manuscript is well written.

Therefore, I suggest publication after addressing the minor comments given below.

L45: Maybe rather state "the air layers above the forest" instead of "free troposphere" as the air will first encounter the roughness sublayer or at nighttime the stable nocturnal boundary layer, then the mixed layer (daytime) or residual layer (nighttime) before reaching the free troposphere.

L96: folia => foliar

L132: To be able to judge potential surface effects and light absorption behavior (cutoff wavelength) of the enclosure please provide information about the material the bags were made of.

L140: Which material were the lines made of?

L147 and Fig. 2: Was there any special reasoning for putting the activated charcoal in front of the Puralfi NOx scrubber? Different to NO2, NO is not well captured by charcoal, and therefore normally the Purafil is put in front of the charcoal as it not only adsorbs NO2, but also oxidizes NO to NO2 which is finally captured by the Purafil and the charcoal. At low ambient NO it might make no difference, but for higher NO this setup normally works better.

L163: Leaf wetness measurements are mentioned here but not presented or discussed in the manuscript. Could you please add these results? Alternatively, at least mention why they were not used.

L168: 1 min zero air measurement plus 2.5 min NO and 2.5 min NO2 results in a 6 min cycle? Please clarify.

L171:To have higher NO2 absorption and less HONO photolysis the light emissions of the diode should be > 390 nm. The 365 nm is close to the HONO absorption band at 368 nm (Stutz et al., 2000). Nevertheless, even at the peak absorption of HONO the absorption cross section of HONO is about a factor of 1.5 smaller and under environmental conditions the HONO to NO2 ratio is normally below 10 % (some cases up to

30 %). Therefore, the HONO interference in ambient air should be small. Due to the high surface to Volume ratio of the chambers, the HONO to NO2 ratio could be higher and might depend on the chamber material. So please also provide information on the chamber material (see above comment).

L300: "Unknown measurement issue for water concentration". As stated in the paper in line 315 the chamber air was not condensing according to the calculated dew points. Could the reason be instead evaporation of surface water films that form at a RH > 50 % due to the deliquescence of deposited salts and other processes (e.g. Burkhardt and Eiden, 1994), Burkhardt and Hunsche, 2013).

L317: Liquid surface films can be formed by processes other than condensation (see esp. Burkhardt and Hunsche, 2013). Furthermore, a recent laboratory chamber study investigated the influence of liquid films (at RH below condensation) on deposition of Peroxyacetyl nitrate and O3 to plants ((Sun et al., 2016) in detail.

L353: As mentioned in the above comments there are processes other than pure condensation that lead to the formation of liquid films. What about the surface wetness measurements? Do they show any changes at RH between 50 and 100 % that could be associated to enhanced O3 uptake?

L361: As the reactions will not stop at the chamber outlet please provide the total residence time from within the chamber (1.5 min) to the analyzer as well?

L365: A very good description of the relevant loss processes for O3 is provided by a recent review (Clifton et al., 2020).

L464: Please revisit this statement in the light of the above comments that liquid films can form at RH> 50 % and therefore contribute to enhanced O3 surface deposition (esp. Sun et al., 2016).

References:

Burkhardt, J. and Eiden, R.: Thin water films on coniferous needles, Atmos. Environ.,

doi:10.1016/1352-2310(94)90469-3, 1994.

Burkhardt, J. and Hunsche, M.: "Breath figures" on leaf surfaces—formation and effects of microscopic leaf wetness, Front. Plant Sci., doi:10.3389/fpls.2013.00422, 2013.

Clifton, O. E., Fiore, A. M., Massman, W. J., Baublitz, C. B., Coyle, M., Emberson, L., Fares, S., Farmer, D. K., Gentine, P., Gerosa, G., Guenther, A. B., Helmig, D., Lombardozzi, D. L., Munger, J. W., Patton, E. G., Pusede, S. E., Schwede, D. B., Silva, S. J., Sörgel, M., Steiner, A. L. and Tai, A. P. K.: Dry Deposition of Ozone Over Land: Processes, Measurement, and Modeling, Rev. Geophys., doi:10.1029/2019RG000670, 2020.

Stutz, J., Kim, E. S. and Platt, U.: UV-visible absorption cross sections of nitrous acid, J. Geophys. Res. VOL. 105, NO. Dll, PAGES , JUNE 16, 2000, 105 (No. D, 14585–14592, 2000.

Sun, S., Moravek, A., Trebs, I., Kesselmeier, J. and Sörgel, M.: Investigation of the influence of liquid surface films on O3 and PAN deposition to plant leaves coated with organic/inorganic solution, J. Geophys. Res., 121(23), doi:10.1002/2016JD025519, 2016.

---

## Referee Comment (RC3) · Anonymous Referee #3 · 28 Apr 2020

1. Does the paper present novel concepts, ideas, tools, or data?

Yes, provides data on set of in-situ trace-gas exchange measurements for important tree species in mixed temperate hardwood forests

2. Are substantial conclusions reached?

Yes, the work demonstrates generally that trace-gas uptake is stomatally controlled, but with some exceptions that imply mesophyll resistance and uptake on cuticles. Shows with in situ measurements that NO2 does not have a measurable compensation point

3. Are the scientific methods and assumptions valid and clearly outlined? -yes

4. Are the results sufficient to support the interpretations and conclusions? yes

5. Is the description of experiments and calculations sufficiently complete and precise to allow their reproduction by fellow scientists (traceability of results)? yes

6. Do the authors give proper credit to related work and clearly indicate their own new/original contribution? yes

7. Does the title clearly reflect the contents of the paper? yes

8. Does the abstract provide a concise and complete summary? yes

9. Is the overall presentation well structured and clear? yes

10. Is the language fluent and precise? Yes, easy to read

11. Are mathematical formulae, symbols, abbreviations, and units correctly defined and used? yes

12. Should any parts of the paper (text, formulae, figures, tables) be clarified, reduced, combined, or eliminated?

Except as noted in comments on points to clarify, no. No suggestions to eliminate anything

13. Are the number and quality of references appropriate? yes

14. Is the amount and quality of supplementary material appropriate? Not applicable

Wang et al present results of a set of in-situ branch enclosure measurements of reactive trace-gas exchange with foliage on trees at the Univ. of Michigan Biological Station. While the time period of the measurements is not very long they represent a significant effort and provide critical confirmation for how trace-gases interact with foliage. I thought the paper was well written and the experimental work was done well. I have some suggestions for clarification of some issues and places that could be expanded a bit to make points more strongly.

Line 96, misspelled foliar

Line 170, What is the conversion efficiency for the photolysis? If it was different from 100% that would show up as a difference in calibration factor for NO and NO2. Those calibrations ought to depend on ambient O3 and light. Can you include some comment on how calibrations depend on ambient condition. I don't doubt you have done it all correctly, but this information might give some additional insight on interpreting the data.

Line 245; The discussion about quantifying the impact of NO2 photolysis could be clarified a little more. Is your point that because the amount of NO emitted is poorly constrained you cannot just compute the uptake of NOx by adding together NO and NO2? Are there enough calibrations at different times of day to examine how the apparent NO2 conversion efficiency varies with light level and account for conversion? What about conversion of NO to NO2 by ambient O3? This ought to be apparent by evaluating the variation in NO calibration constant as a function of O3

Line 310, Can you go the next step after concluding that there is some mesophyll resistance? The effect of having a non-zero mesophyllic resistance is a non-zero concentration inside the leaf. Using equation 2 and 3 you could solve for NO2 concentrations internal to leaf. Similarly, for situations with excess deposition you could compute a value for cuticular deposition from the residual after subtracting the stomatal uptake. Granted you wouldn't get a unique solution if there were both cuticular uptake and non-stomatal deposition. But you can make this section stronger by quantifying some values for the other processes you point to. At the end of paragraph, having some values for range of mesophyllic conductance would be better than just stating further investigation is needed.

Line 318 In addition to there being a possibility cuticular adsorption accounts for extra NO2 deposition you might also note that stomatal enclosure might not be complete. Discussion about whether stomatal conductance goes to zero shows up mostly in discussions seeking to explain sap flow or water flux that doesn't go to zero at night. It might not be as much of an issue for daytime periods, but could be noted just for completeness. Can you also comment on how much of the data are for conditions that the vapor pressure differences between leaf and ambient air are quite small so that stomatal conductance computation has larger uncertainty. At the limit when ambient air approached saturation and leaf and air temperature were equal the stomatal conductance couldn't be determined from water flux. You have noted in that leaf temperature always exceeded dewpoint in the context of discounting possibility of dew, but it is also relevant for evaluating how well stomatal conductance is defined.

Can you comment more on O3 uptake. I agree that reaction with VOC could be an important loss process for O3 in addition to reaction with foliage. Your point would be stronger, however, by providing the rate constants for the VOC typically associated with white pine not just pointing out one with the highest reaction rate as well as noting that the oak and aspen are know isoprene emitters.

Line 392 Say something more about the difference in chamber temperature for the aspen branch compared to the other species. Is this because the outside air temperature was cooler also, or on account differences in radiation? Large differences in conditions between the species make comparisons among them more difficult. Can you say anything about whether the chamber conditions relative to outside conditions were different for the species. Finally, the point that aspen have stomata on both sides of leaf ought to come first, and rather than speculate that they have more stomata per unit area find some data in the literature about this. There is no need for several lines of explanation about why the conductance is higher before this point about double sided stomata. The explanation only needs to explain the additional enhancement beyond twice. Likewise the other differences should come first. Does the extra water flux account for reduced temperatures in apsen enclosures?

Line 510: It would be preferable to have data availability point to an existing data set already available rather than just making it available on request. It easier for the investigator to just prepare the files once and submit to a data server (doesn't UMBS have this for work at the site). Additionally, if the investigators move or retire then data on request gets hard to find years from now.
* * *

---

## Author Comment (AC1) · 18 Jul 2020

**Author response to Anonymous Referee #1 of ACP-2020-149, "Measurement report: Leaf-scale gas exchange of atmospheric reactive trace species (NO2, NO, O3) at a northern hardwood forest in Michigan"**

We greatly appreciate the thoughtful feedback provided by Anonymous Referee #1. The questions and comments have helped to improve and enhance the manuscript. Below, we address each comment individually. Referee comments are given in **Bold**, author responses are given in normal font, changes to make in the manuscript are given in blue.

**P2 L60: The citation of Delaria et al., 2018 and reported estimation of 15-30% removal of soil-emitted NOx is correctˇAˇToak woodlands have a very low LAI. However, in Delaria and Cohen 2020 (now published and not in discussion), they report much larger canopy reductions for forests with more typical LAI, in line with the 25-55% loss previously reported.**

The latest publication of Delaria and Cohen (2020) has been cited.

In the manuscript, P2:
"Implementing these results in a multi-layer single-column model, it was calculated that California oak woodland canopy removes 15-30% of soil-emitted $NO_x$, and other forests in California and Michigan, close to 60% (Delaria and Cohen, 2020)."

**P3 L64: Extra parenthesis**

It is removed.

**P3 L83: Would be nice if the instrument were stated explicitly.**

That information has been added to the text on P3. It was a chemiluminescent NOx detector equipped with a highly $NO_2$ specific blue light converter.

**P3 L96: Correct "folia" to "foliar"**

Corrected.

**P7 L205: Several studies have observed significant stomatal opening during the night (e.g. Dawson et al., 2007–10.1093/treephys/27.4.561). Consider adding a discussion of how, if this was occurring in your chamber, this assumption would have affected your results (if at all).**

We have modified the manuscript text to address this point:

In the manuscript, P7:
"Nighttime transpiration in trees and shrubs has been measured in prior work, with reports of nighttime transpiration rates ranging from 0 to as much as 25% of the daytime value (Dawson et al., 2007), suggesting that leaf stomata may remain open at night for some plants. However, this possibility did not affect the above results as there was no evidence of a consistent concentration difference above zero between the enclosure outlet and inlet measurements."

**P8 L246: A more detailed description of your empty chamber photolysis corrections would be useful.**

We have modified the paragraph to give a more specific and detailed description on the corrections based on the empty chamber measurements.

(Please also see response to RC#3, L245.)

In the manuscript, P8:

"There are a couple of factors that complicated the $NO_2$ gas exchange experiment. First, the $NO_2$ standard used for delivering $NO_2$ to the enclosure contains about 5% NO that was unavoidably added to the enclosure. Secondly, when there was intense direct sunlight, some $NO_2$ in the enclosure was photolyzed. While corrections for these interferences were done using the measurements from the reference enclosure, it is difficult to completely remove the artifact caused by $NO_2$ photolysis. This is because the sunlight exposure of the two enclosures, although situated side-by-side, was often uneven, and the measurements of the enclosures were done not simultaneously but sequentially. This problem is particularly pronounced for clear sky conditions with strong contrasts in sunlit and shaded conditions inside the canopy. The branch enclosure was always positioned to get more sun exposure than the reference enclosure if choices needed to be made. Therefore, the branch enclosure likely received more sunlight overall, even though it might be more shaded during some measurement cycles. Generally, for the periods of strong sunlight, there is residual NO after the correction against the reference enclosure is made. If we assume all this is due to an underestimation of $NO_2$ photolysis and make a further correction by combining the changes of NO and $NO_2$, the data quality is not improved while more noise is introduced to the data. Because of this and because we are not absolutely certain about all possible sources of $NO_x$ from the branch enclosures, we prefer to adhere with the correction using only the reference enclosure measurements and view the resulting $NO_2$ flux as an upper bound, with possibly as much as 20% overestimation under direct sunlight conditions, which accounts for ~16% of all data during the $NO_2$ exchange experiments."

**P10 L314: Units for the intercept should be added. Additionally, under the resistance model framework you discuss, the relationship of Vd to gH2O is non-linear. How might this affect your inferences of cuticular uptake?**

We have added the units (mm s$^{-1}$) to the intercept.

For the second question:
In the resistance model framework at the leaf scale, the cuticular, stomatal, and mesophyll resistances are the main factors to determine the deposition velocity. Putting them together, the overall conductance (i.e. the inverse of the total resistance) to deposition is:

$1/R_{total} = 1/R_{cut} + 1/(R_{sto} + R_{meso})$.

Thus, if $R_{cut} \gg (R_{sto} + R_{meso})$ or $R_{cut}$ is constant and $R_{meso} \ll R_{sto}$, the deposition velocity vs. stomatal conductance will be linear. If $R_{meso}$ is significant compared to $R_{sto}$, the deposition velocity will be limited by $R_{meso}$ as stomatal conductance increases. Of course, we don't know if and how $R_{cut}$ and $R_{meso}$ vary with environmental factors, which would potentially complicate the $V_d$ to $g_{H2O}$ relationship.

We added the following in the manuscript, P14:

"When extrapolated to zero stomatal conductance, the deposition velocity of $NO_2$ to white pine was 0.43 mm s$^{-1}$ (Figure 6a), implying deposition unrelated to leaf stomata, possibly to wet leaf surfaces and/or to leaf cuticula. This observation does not exclude the possible existence of these pathways when the stomata are open. A deposition velocity higher than expected based on the stomatal conductance would result if there is significant non-stomatal deposition. On the other hand, mesophyll resistance renders a lower deposition velocity than the expected value. There is no mechanistic reason why the deposition velocity associated with either a non-stomatal pathway or mesophyll resistance should remain constant or vary linearly with stomatal conductance. The relationship of deposition velocity, $v_{d\_NO_2}$, and stomatal conductance, $g_{H_2O}$, would remain essentially linear as long as stomatal deposition dominates or the non-stomatal deposition term is constant while mesophyll resistance is small. However, if mesophyll resistance is significant, it would limit the increase of $v_{d\_NO_2}$ with stomatal conductance. "

**P12 L363: How high of emission rates would this require? Is it outside the range reported for trees of the species considered?**

Yes, it does require an unreasonably high emission rate to match the amount of ozone removed.

Prompted by this comment, we have modified this argument using literature values of BVOC emission rates and speciation to give a more realistic estimation of the amount of ozone loss due to chemical reactions. It is < 1%.

(Please also see  RC#3, p 3-4, "Can you comment more on O3 uptake…".)

In the manuscript, P12:

"Estimation of the possible contribution from gas-phase reactions with BVOCs was made as follows. The upper bounds of typical emission rates at 30°C and PAR level at 1000 µmol m$^{-2}$ s$^{-1}$ for monoterpenes and other BVOCs (excluding isoprene) are 3 and 5 µg C g$^{-1}$ h$^{-1}$, respectively (Guenther et al., 1994). The speciation of major BVOCs emitted by white pine at UMBS is based on Kim et al. (2011), including $\alpha$- and $\beta$-pinene, limonene, linalool, $\alpha$-humulene, and $\beta$-caryophyllene. Using the rate constants of the BVOCs with ozone reactions (Burkholder et al., 2015), and the residence time of 1.5 min in the enclosure plus ~6 sec in the sample line before reaching the detector, the estimated ozone loss due to gas-phase chemical reactions was less than 1%. Even with optimal light and temperature conditions for BVOC emission, the estimated gas-phase chemical removal would only be on the order of a few percent."

**P13 L395-396: should this be VPD?**

Yes. It is fixed now.

**P13 L403: "In the future,"? "In future work,"?**

Suggestion is taken.

**P14 L437: This was also a conclusion of Delaria et al., 2018.**

We added this information in the text.

In the manuscript, P15:

"Delaria et al (2018) reached the same conclusion from their study on *Quercus agrifolia*."

**O3 deposition: There are a number of recent references discussing ozone deposition that are not included. The paper would be stronger if it placed itself in the context of these and other recent papers on the subject (e.g. Silva and Heald GRL 2018 https://doi.org/10.1002/2017JD027278, Kavassalis and Murphy GRL 2017 https://doi.org/10.1002/2016GL071791 and Clifton et al. in Reviews of Geophysics 2020 https://doi.org/10.1029/2019RG000670).**

We took the suggestion and added the references and relevant context in the manuscript.

In the manuscript, P2:

"Similarly, vegetation and plant surfaces also affect ozone levels through dry deposition (Clifton et al., 2019, 2020; Kavassalis and Murphy, 2017; Silva and Heald, 2018). In forested areas, ozone dry deposition occurs through leaf stomata as well as non-stomatal pathways including cuticular uptake, and wet or dry leaf surface reactions, while some $O_3$ is also removed by gas-phase chemical reactions e.g. with biogenic volatile organic compounds (BVOCs) and NO. Though these processes have been identified, the exact partitioning between the dry deposition pathways (and in-canopy chemical destruction) has not been unequivocally determined, hindering the ability to correctly assess ground-level ozone."

---

## Author Comment (AC2) · 18 Jul 2020

**Author response to** Anonymous Referee #2 **of ACP-2020-149: "Measurement report: Leaf-scale gas exchange of atmospheric reactive trace species (NO2, NO, O3) at a northern hardwood forest in Michigan"**

We greatly appreciate the thoughtful feedback provided by Anonymous Referee #2. The questions and comments have helped to improve and enhance the manuscript. Below, we address each comment individually. Referee comments are given in **Bold**, author responses are given in normal font, changes made to the text in the manuscript are given in blue. The revised manuscript includes all the changes listed below.

**L45: Maybe rather state "the air layers above the forest" instead of "free troposphere" as the air will first encounter the roughness sublayer or at nighttime the stable nocturnal boundary layer, then the mixed layer (daytime) or residual layer (nighttime) before reaching the free troposphere.**

We have changed the phrase and now the text (page 2) reads:

"The relative differences in the time scales of the turbulent mixing and the chemical and physical sink processes determine the amount of $NO_x$ removed within the canopy, with the remaining $NO_x$ being released into the boundary layer."

**L96: folia => foliar**

It's corrected.

**L132: To be able to judge potential surface effects and light absorption behavior (cutoff wavelength) of the enclosure please provide information about the material the bags were made of.**

The material was Tedlar, polyvinyl fluoride. This info is now in the manuscript.

**L140: Which material were the lines made of?**

It's polytetrafluoroethylene (PTFE), also added to the manuscript.

**L147 and Fig. 2: Was there any special reasoning for putting the activated charcoal in front of the Puralfi NOx scrubber? Different to NO2, NO is not well captured by charcoal, and therefore normally the Purafil is put in front of the charcoal as it not only adsorbs NO2, but also oxidizes NO to NO2 which is finally captured by the Purafil and the charcoal. At low ambient NO it might make no difference, but for higher NO this setup normally works better.**

There was no specific reason to put the Charcoal before Purafil. Fortunately, during our work, the ambient NO concentrations were generally low in the relatively remote forest. Thank you very much for the information!

**L163: Leaf wetness measurements are mentioned here but not presented or discussed in the manuscript. Could you please add these results? Alternatively, at least mention why they were not used.**

Thank you for this point. We only used the leaf wetness results for qualitative/diagnostic purposes. Therefore, it was not mentioned in the data analysis. We added a short explanation in the text.

In the manuscript, P6:

"leaf wetness (for qualitative assessment of leaf conditions only) (S-LWA, Onset)"

**L168: 1 min zero air measurement plus 2.5 min NO and 2.5 min NO2 results in a 6 min cycle? Please clarify.**

It was a mistake. It should be "1 min zero air measurement plus 2 min NO and 2 min $NO_2$". Thank you for catching it.

**L171: To have higher NO2 absorption and less HONO photolysis the light emissions of the diode should be > 390 nm. The 365 nm is close to the HONO absorption band at 368 nm (Stutz et al., 2000). Nevertheless, even at the peak absorption of HONO the absorption cross section of HONO is about a factor of 1.5 smaller and under environmental conditions the HONO to NO2 ratio is normally below 10 % (some cases up to 30 %). Therefore, the HONO interference in ambient air should be small. Due to the high surface to Volume ratio of the chambers, the HONO to NO2 ratio could be higher and might depend on the chamber material. So please also provide information on the chamber material (see above comment).**

We apologize for this mistake. We used Hamamatsu L11921-500 LED light sources. The peak wavelength was 385+/-5 nm.

**L300: "Unknown measurement issue for water concentration". As stated in the paper in line 315 the chamber air was not condensing according to the calculated dew points. Could the reason be instead evaporation of surface water films that form at a RH > 50% due to the deliquescence of deposited salts and other processes (e.g. Burkhardt and Eiden, 1994), Burkhardt and Hunsche, 2013).**

Please see the response after the comment for L353.

**L317: Liquid surface films can be formed by processes other than condensation (see esp. Burkhardt and Hunsche, 2013). Furthermore, a recent laboratory chamber study investigated the influence of liquid films (at RH below condensation) on deposition of Peroxyacetyl nitrate and O3 to plants ((Sun et al., 2016) in detail.**

Please see the response below.

**L353: As mentioned in the above comments there are processes other than pure condensation that lead to the formation of liquid films. What about the surface wetness measurements? Do they show any changes at RH between 50 and 100 % that could be associated to enhanced O3 uptake?**

Regarding the last three comments, we thank the reviewer for bringing this to our attention. We do not have observational results to address the point about microscopic water film but it certainly is an aspect to investigate in future experiments. We have modified the text to reflect the possibility of microscopic

water film on the leaf surface. The leaf wetness sensor likely did not have similar deposits to aid the formation of water film because the sensor was cleaned before being placed in the enclosure. (The leaves were not). The wetness sensor showed relatively dry conditions, i. e. lower readings during the $O_3$ experiment compared to those at early morning hours, and the RH was between 70-80%.

We modified the manuscript text to reflect this possibility. On page 14:

"To assess the role of wet leaf surfaces to non-stomatal deposition, we calculated the white pine enclosure dew point using the temperature and relative humidity data and compared it to the measured leaf temperature. The leaf temperature was always higher than the dew point during the experiments, excluding the possibility of a wet leaf surface from the condensation of pure water. However, a microscopic water film may nevertheless form at a relative humidity as low as 50% if there are hygroscopic deposits on the leaf surface (Burkhardt and Eiden, 1994; Burkhardt and Hunsche, 2013; Sun, 2016). The microscopic water film could potentially modify gas exchange rates of water-soluble trace gases in the air. Data from this work do not contain information that can be used to delineate the possibilities of trace gas dissolution into microscopic water films or cuticular uptake. Further investigations with appropriately designed experiments, better measurement precisions, longer observation time, and under different environmental conditions are necessary to delineate the various possible deposition pathways and their dependencies."

**L361: As the reactions will not stop at the chamber outlet please provide the total residence time from within the chamber (1.5 min) to the analyzer as well?**

The residence time in the sample line was about 6 seconds. (1/4" OD, 1/8" ID, 100 feet). This info is now added to the manuscript.

**L365: A very good description of the relevant loss processes for O3 is provided by a recent review (Clifton et al., 2020).**

Yes. We have now cited this reference in the revised text.

**L464: Please revisit this statement in the light of the above comments that liquid films can form at RH> 50 % and therefore contribute to enhanced O3 surface deposition (esp. Sun et al., 2016).**

Yes, this statement was modified to include the possibility of a thin water film at low RH and above the dew point.

In the manuscript, P16:

"A brief survey of the foliar $O_3$ loss found that uptake by the deciduous trees also closely followed stomatal conductance, while the $O_3$ foliar deposition velocity for white pine was much larger than expected from leaf stomatal uptake alone. Removal via gas-phase chemical reactions was calculated to be negligible based on estimates of known BVOC emission rates and speciation, implying other non-stomatal pathways - cuticular uptake, dissolution to wet leaf surfaces, and/or chemical reactions at the leaf surface – are responsible for the additional ozone deposition, with their relative importance to be determined. "

---

## Author Comment (AC3) · 18 Jul 2020

**Author response to Anonymous Referee #3 of ACP-2020-149: "Measurement report: Leaf-scale gas exchange of atmospheric reactive trace species (NO2, NO, O3) at a northern hardwood forest in Michigan"**

We greatly appreciate the thoughtful feedback provided by Anonymous Referee #3. The questions and comments have helped to improve and enhance the manuscript. Below, we address each comment individually. Referee comments are given in **Bold**, author responses are given in normal font, changes made to the text in the manuscript are given in blue. The revised manuscript includes all the changes listed below.

**Line 96, misspelled foliar**

It's corrected.

**Line 170, What is the conversion efficiency for the photolysis? If it was different from 100% that would show up as a difference in calibration factor for NO and NO2. Those calibrations ought to depend on ambient O3 and light. Can you include some comment on how calibrations depend on ambient condition. I don't doubt you have done it all correctly, but this information might give some additional insight on interpreting the data.**

The reactive trace gases in ambient air were scrubbed prior to the purge air entering the enclosure. So, measurements of the $NO_x$ added to the enclosure were not subject to $O_3$ interference.

For the calibrations, UHP zero air, an NO standard, and UHP $O_2$ (for making $O_3$) were used. These steps did not involve ambient air.

During the calibrations, NO was calibrated directly using an NO standard diluted with the UHP zero air. For the $NO_2$ calibration, the same diluted NO standard was titrated with some $O_3$ to form a mixture of $NO_2$ and NO, then the NO was measured, which gave **the expected $NO_2$** counts ($NO_2$ = NO_original_amount – NO_after_adding_O₃). Afterward, $NO_2$ was photolyzed to NO using the LED, and the total NO was measured, yielding **the measured $NO_2$** counts. The ratio: (**measured_$NO_2$ /expected_$NO_2$**), is the conversion efficiency. It was fairly steady throughout the campaign at ~0.68. The instrument was calibrated every 7 hours.

We added in the manuscript, P6:

"…with a conversion efficiency of ~0.68. Because the ambient air was scrubbed to remove $O_3$ (and other trace gases) before entering the enclosures, the effect of ambient $O_3$ on $NO_x$ measurements was negligible."

**Line 245; The discussion about quantifying the impact of NO2 photolysis could be clarified a little more. Is your point that because the amount of NO emitted is poorly constrained you cannot just compute the uptake of NOx by adding together NO and NO2? Are there enough calibrations at different times of day to examine how the apparent NO2 conversion efficiency varies with light level and account for conversion? What about conversion of NO to NO2 by ambient O3? This ought to be apparent by evaluating the variation in NO calibration constant as a function of $O_3$.**

We have modified the text to explain the background correction using the empty enclosure better. Regarding photolysis, the two enclosures could not be measured simultaneously with one instrument, and they were in mid-canopy instead of under direct sunlight without branches above. So, the light exposure of the two chambers was not identical, however close, making the correction of $NO_2$ photolysis an approximation.

Conversion of $NO_2$ to $NO$ was done in the instrument using an LED light source. The calibration was done every 7 hours.

The ambient air was scrubbed free from $O_3$ before it was sent to the enclosures. If there's any residual $O_3$, it's below the detection limit of the $O_3$ detector (<0.5 ppb). This is not enough $O_3$ to significantly convert $NO$ to $NO_2$ within the residence time in the enclosure and the sample line (total ~96 sec).

(Please also see response to RC#1, P8 L246.)

The text in the manuscript is modified as follows, on page 8:

"There are a couple of factors that complicated the $NO_2$ gas exchange experiment. First, the $NO_2$ standard used for delivering $NO_2$ to the enclosure contains about 5% $NO$ that was unavoidably added to the enclosure. Secondly, when there was intense direct sunlight, some $NO_2$ in the enclosure was photolyzed. While corrections for these interferences were done using the measurements from the reference enclosure, it is difficult to perfectly remove the artifact caused by $NO_2$ photolysis. This is because the sunlight exposure of the two enclosures, although situated side-by-side, was often uneven, and the measurements of the enclosures were done not simultaneously but sequentially. This problem is particularly pronounced when it was cloudless with a strong contrast of light and shade inside the canopy. The branch enclosure was always placed for better sun exposure than the reference enclosure if choices needed to be made. Therefore, the branch enclosure likely received more sunlight overall, even though it might be more shaded during some measurement cycles. Generally, for the periods of strong sunlight, there is residual $NO$ after the correction against the reference enclosure is made. If we assume all this is due to an underestimation of $NO_2$ photolysis and make a further correction by combining the changes of $NO$ and $NO_2$, the data quality is not improved while more noise is introduced to the data. Because of this and because we are not absolutely certain about all possible sources of $NO_x$ from the branch enclosures, we prefer to adhere with the correction using only the reference enclosure measurements and view the resulting $NO_2$ flux as an upper bound, with possibly as much as 20% overestimation under direct sunlight conditions, which accounts for ~16% of all data during the $NO_2$ exchange experiments. "

**Line 310, Can you go the next step after concluding that there is some mesophyll resistance? The effect of having a non-zero mesophyllic resistance is a non-zero concentration inside the leaf. Using equation 2 and 3 you could solve for NO2 concentrations internal to leaf. Similarly, for situations with excess deposition you could compute a value for cuticular deposition from the residual after subtracting the stomatal uptake.**

**Granted you wouldn't get a unique solution if there were both cuticular uptake and non-stomatal deposition. But you can make this section stronger by quantifying some values for the other processes**

**you point to. At the end of paragraph, having some values for range of mesophyllic conductance would be better than just stating further investigation is needed.**

The concentration of $NO_2$ in the leaf internal air space indeed can be calculated using equations 2 and 3. However, I am not sure if the numbers can be put into a meaningful context because the mesophyll resistance was inferred from data where the deposition rate was lower than the theoretical value from $g_{H2O}$, but not systematically investigated. Also, if both cuticular uptake and mesophyll resistance exist, their effects would be in the opposite direction. $NO_2$ goes through disproportionation when it dissolves in water, thus the $NO_2$ pressure in the internal air space needs to be in steady state with the solution. These make it difficult to interpret the apparent internal $NO_2$ pressure.

**Line 318 In addition to there being a possibility cuticular adsorption accounts for extra NO2 deposition you might also note that stomatal enclosure might not be complete. Discussion about whether stomatal conductance goes to zero shows up mostly in discussions seeking to explain sap flow or water flux that doesn't go to zero at night. It might not be as much of an issue for daytime periods, but could be noted just for completeness.**

We did observe some water flux at nighttime, especially in the white pine enclosure. Correspondingly there was nighttime deposition of $NO_2$ to pine needles. The additional deposition attributed to cuticular or other non-stomatal processes was inferred by extrapolating the deposition velocity to zero stomatal conductance. The nighttime stomatal conductance of other trees was very small or close to zero.

**Can you also comment on how much of the data are for conditions that the vapor pressure differences between leaf and ambient air are quite small so that stomatal conductance computation has larger uncertainty. At the limit when ambient air approached saturation and leaf and air temperature were equal the stomatal conductance couldn't be determined from water flux. You have noted in that leaf temperature always exceeded dewpoint in the context of discounting possibility of dew, but it is also relevant for evaluating how well stomatal conductance is defined.**

About 10% of the data show small VPD (< 1 standard deviation from zero) under the conditions you mentioned. It usually happened in the early morning just before and around sunrise. The trace gas fluxes were relatively small during these times.

The above is added to the manuscript, P9:

"When the conditions are such that the difference between the leaf and air temperatures is small and the enclosure humidity is high, the difference between $C_{H_2O\_leaf}$ and $C_{H_2O_{enclosure}}$ is also reduced, increasing the uncertainty in $g_{H_2O}$ . In our measurements, this happened mostly from dawn to sunrise, accounting ~10% of the total data points, where the $(C_{H_2O\_leaf} - C_{H_2O\_enclosure})$ was within one standard deviation from zero. "

**Can you comment more on O3 uptake. I agree that reaction with VOC could be an important loss process for O3 in addition to reaction with foliage. Your point would be stronger, however, by providing the rate constants for the VOC typically associated with white pine not just pointing out one with the highest reaction rate as well as noting that the oak and aspen are known isoprene emitters.**

The part regarding gas-phase reactions has been modified to use emission rates and speciation from literature to give a more realistic estimation.

(Please also see RC#1 P12 L363.)

In the manuscript, page 12:

"Estimation of the possible contribution from gas-phase reactions with BVOCs was made as follows. The upper bounds of typical emission rates at 30°C and PAR level at 1000 $\mu$mol m$^{-2}$ s$^{-1}$ for monoterpenes and other BVOCs (excluding isoprene) are 3 and 5 $\mu$g C g$^{-1}$ h$^{-1}$, respectively (Guenther et al., 1994). The speciation of major BVOCs emitted by white pine at UMBS is based on Kim et al. (2011), including $\alpha$- and $\beta$-pinene, limonene, linalool, $\alpha$-humulene, and $\beta$-caryophyllene. Using the rate constants of the BVOCs with ozone reactions (Burkholder et al., 2015), and the residence time of 1.5 min in the enclosure plus ~6 sec in the sample line before reaching the detector, the estimated ozone loss due to gas-phase chemical reactions was less than 1%. Even with optimal light and temperature conditions for BVOC emission, the estimated gas-phase chemical removal would only be on the order of a few percent."

**Line 392 Say something more about the difference in chamber temperature for the aspen branch compared to the other species. Is this because the outside air temperature was cooler also, or on account differences in radiation? Large differences in conditions between the species make comparisons among them more difficult. Can you say anything about whether the chamber conditions relative to outside conditions were different for the species. Finally, the point that aspen have stomata on both sides of leaf ought to come first, and rather than speculate that they have more stomata per unit area find some data in the literature about this. There is no need for several lines of explanation about why the conductance is higher before this point about double sided stomata. The explanation only needs to explain the additional enhancement beyond twice. Likewise the other differences should come first. Does the extra water flux account for reduced temperatures in apsen enclosures?**

The text has been modified based on the suggestions. The daily *average* temperatures inside and outside the chamber were similar. The air temperature was cooler on the days when aspen was sampled. The VPD during this time was smaller than in other enclosures when it was warmer.

We could not find in the literature specific records on the stoma distribution of aspen except populus in general. The part regarding stomata on both sides of aspen leaves was moved to the beginning of this part of the discussion as suggested.

In the manuscript, P13:

"Biological features may have contributed to this difference. Compared with the other three trees in this work, the aspen was younger and smaller. The enclosed branch was in the upper part of the crown containing developing new leaves. Past measurements, albeit on different species, have shown that for the same species under similar environmental conditions, leaves of young trees generally have higher stomatal conductance than old ones (Fredericksen et al., 1995; Hubbard et al., 1999; Niinemets, 2002; Yoder et al., 1994). Another possible reason for the observed high $g_{H_2O}$ , while direct evidence has yet to be found, is the number of stomata. Many trees have stomata on only the lower (abaxial) leaf surface; however, trees that belong to the genus Populus, which includes aspen, are an exception. They have stomata on both sides (amphistomatous), a feature that allows increased photosynthetic rate and fast growth (Kirkham, 2014). If the bigtooth aspen leaves are indeed amphistomatous, a relatively high

$g_{H_2O}$ can be expected. We compared the environmental conditions of the enclosures. The integrated PAR exposure levels were similar. The daily variation of the relative humidity in the bigtooth aspen enclosure was not significantly different from the others. In contrast, the average daily temperature was 19.2°C, cooler than the temperatures (23.9°C, 22.6°C, and 21.6°C) in the other enclosures, similar to the average ambient air temperature outside the enclosure during the same time, 19.1°C, and 23.6°C, 22.4°C, and 21.3°C. The combined conditions of moisture and temperature led to a relatively low vapor pressure deficit (VPD) in the aspen enclosure, 0.8 kPa, compared to 1.2 kPa (white pine), 1.0 kPa (red maple), and 1.4 kPa (red oak) in the others. Generally, VPD and $g_{H2O}$ are inversely correlated and a low VPD corresponds to a relatively high $g_{H2O}$ (Hubbard et al., 1999; Urban et al., 2017a, 2017b). However, because here we are comparing different tree species, we consider the observed results to stem from the combination of biological and environmental factors. Further examination of these factors is beyond the scope of this paper, nevertheless, it would be beneficial to take this temporal and spatial variability and inhomogeneity into account in model parameterizations of trace gas dynamics since plant stomata are the main conduit of $NO_x$ and $O_3$ deposition over vegetation."

**Line 510: It would be preferable to have data availability point to an existing data set already available rather than just making it available on request. It easier for the investigator to just prepare the files once and submit to a data server (doesn't UMBS have this for work at the site). Additionally, if the investigators move or retire then data on request gets hard to find years from now.**

Yes, the data will be archived at https://umich.box.com/v/PROPHETAMOS2016.

---

## Editor Decision (ED1)

1) Your response to the following referee comments was quite detailed. However, it might be worthwhile adding some of this information to the manuscript.

*Original referee comment:*
*Line 318 In addition to there being a possibility cuticular adsorption accounts for extra*
*$NO_2$ deposition you might also note that stomatal enclosure might not be complete.*
*Discussion about whether stomatal conductance goes to zero shows up mostly in discussions seeking to explain sap flow or water flux that doesn't go to zero at night. It might not be as much of an issue for daytime periods, but could be noted just for completeness.*

2) l. 398ff and Eq.-1: For the calculations using Eq-1, you assume a single-sided leaf area. How does the fact that aspen leaves might have stomata on both sides may affect results based on this assumption? Do the findings as described in l. 440 possibly point to the same phenomenon?

**Technical comments**

All equations: Add units to all parameters, either in the equations or in the text where the parameters are defined

l. 13: there seems to be a word missing (compensation point of …?)

l. 43, 44; 58, 62, 89, 249ff; l. 487 (and possibly at other places): check all chemical formulae and correct indices where necessary

l. 102 folia → foliar

l. 124: replace 'see below' by reference to specific section

l. 179: 'conversion' misspelled

l. 234: appears → appear

l. 277: add 'for' (accounting for…)

l. 315; l. 330: refer to specific subsections in the discussion section

l. 317: Reword 'the same slope is 0.4'

l. 333: Reword 'the stomatal conductance of each data point…'

l. 393: 'Biological features' sounds very general. Can you be more specific?

---

## Author Response (AR2)

1) Your response to the following referee comments was quite detailed. However, it might be worthwhile adding some of this information to the manuscript.

**Original referee comment:**

Line 318 In addition to there being a possibility cuticular adsorption accounts for extra NO2 deposition you might also note that stomatal enclosure might not be complete. Discussion about whether stomatal conductance goes to zero shows up mostly in discussions seeking to explain sap flow or water flux that doesn't go to zero at night. It might not be as much of an issue for daytime periods, but could be noted just for completeness.

**On page 11, 2nd paragraph, we added the following:**

"The nighttime stomatal conductance of white pine is relatively high, with a median value at 0.57 mm s-1, compared to 0.05-0.19 mm s-1 for the other trees, probably due to incomplete stomatal closure at night (Dawson et al., 2007). There is corresponding nighttime deposition of NO2, with a higher rate for white pine relative to the other trees (Fig. 5)."

**2) I. 398ff and Eq.-1: For the calculations using Eq-1, you assume a single-sided leaf area. How does the fact that aspen leaves might have stomata on both sides may affect results based on this assumption? Do the findings as described in I. 440 possibly point to the same phenomenon?**

Yes, we used the single-sided leaf area to calculate the flux, and the findings described in I. 440 include the possible effect of having stomata on both sides of the leaf. If aspen indeed has stomata on both sides of its leaves, then the fluxes with respect to stomata-containing leaf area would be half of the reported values. However, for the purpose of comparing different trees, it is more straightforward to consistently count the leaf area as all single-sided or double-sided.

**Technical comments**

**All equations: Add units to all parameters, either in the equations or in the text where the parameters are defined**

The units are added. For parameters that may have different units due to the scale of the quantity, e.g. pmol vs. mmol, we chose one of them (in this case, pmol) to list. Please advise if it should be done differently.

**I. 13: there seems to be a word missing (compensation point of ...?)**

The word "of" is deleted.

**I. 43, 44; 58, 62, 89, 249ff; I. 487 (and possibly at other places): check all chemical formulae and correct indices where necessary**

We did a global search and check.

**I. 102 folia $\rightarrow$ foliar**

The letter "r" is added.

**I. 124: replace 'see below' by reference to specific section**

It's changed to "see Methods section".

**I. 179: 'conversion' misspelled**

It's corrected.

**I. 234: appears $\rightarrow$ appear**

It's changed to "appear".

**I. 277: add 'for' (accounting for...)**

"For" is added.

**I. 315; I. 330: refer to specific subsections in the discussion section**

We changed the text from "Discussion section" to "Discussion section 4.1".

**I. 317: Reword 'the same slope is 0.4'**

We changed "the same slope is 0.4" to "the slope of  $v_{dNO_2}$  vs.  $g_{H_2O}$  is 0.4".

**I. 333: Reword 'the stomatal conductance of each data point...'**

We changed the above text to "The amplitude of the stomatal conductance for each data point...".

**I. 393: 'Biological features' sounds very general. Can you be more specific?**

We changed "Biological features" to "Biological features, such as plant and leaf age, stomatal density, ...".

**Measurement report: Leaf-scale gas exchange of atmospheric reactive trace species (NO2, NO, O3) at a northern hardwood forest in Michigan**

Wei Wang1, Laurens Ganzeveld2, Samuel Rossabi1, Jacques Hueber1, Detlev Helmig1

1Institute of Arctic and Alpine Research, University of Colorado, Boulder, CO 80309, USA 2Meteorology and Air Quality, Wageningen University, 6708 PB, Netherlands

Correspondence-to: Wei Wang (wei.wang-3@Colorado.edu)

5

Abstract. During the Program for Research on Oxidants: PHotochemistry, Emissions, and Transport (PROPHET) campaign from July 21 to August 3, 2016, field experiments of leaf-level trace gas exchange of nitric oxide (NO), nitrogen dioxide

- 10 (NO2), and ozone (O3) were conducted for the first time on the native American tree species *Pinus strobus* (eastern white pine), *Acer rubrum* (red maple), *Populus grandidentata* (bigtooth aspen), and *Quercus rubra* (red oak) in a temperate hardwood forest in Michigan, USA. We measured the leaf-level trace gas exchange rates and investigated the existence of an NO2 compensation point-of, hypothesized based on a comparison of a previously observed average diurnal cycle of NOx (NO2 + NO) concentrations with that simulated using a multi-layer canopy exchange model. Known amounts of trace gases were
- 15 introduced into a tree branch enclosure and a paired blank reference enclosure. The trace gas concentrations before and after the enclosures were measured, as well as the enclosed leaf area (single-sided) and gas flow rate to obtain the trace gas fluxes with respect to leaf surface. There was no detectable NO uptake for all tree types. The foliar NO2 and O3 uptake largely followed a diurnal cycle, correlating with that of the leaf stomatal conductance. NO2 and O3 fluxes were driven by their concentration gradient from ambient to leaf internal space. The NO2 loss rate at the leaf surface, equivalently, the foliar NO2
- 20 deposition velocity toward the leaf surface, ranged  $0-3.6 \text{ mm s}^{-1}$  for bigtooth aspen, and  $0-0.76 \text{ 
[revised manuscript text omitted]